# Single-cell transcriptomics following ischemic injury identifies a role for B2M in cardiac repair

Bas Molenaar[1,3], Louk T. Timmer [1,3], Marjolein Droog[1], Ilaria Perini[1], Danielle Versteeg[1,2], Lieneke Kooijman[1], Jantine Monshouwer-Kloots[1], Hesther de Ruiter[1], Monika M. Gladka [1] & Eva van Rooij [1,2 ✉]

The efficiency of the repair process following ischemic cardiac injury is a crucial determinant for the progression into heart failure and is controlled by both intra- and intercellular signaling within the heart. An enhanced understanding of this complex interplay will enable better exploitation of these mechanisms for therapeutic use. We used single-cell transcriptomics to collect gene expression data of all main cardiac cell types at different time-points after ischemic injury. These data unveiled cellular and transcriptional heterogeneity and changes in cellular function during cardiac remodeling. Furthermore, we established potential inter-cellular communication networks after ischemic injury. Follow up experiments confirmed that cardiomyocytes express and secrete elevated levels of beta-2 microglobulin in response to ischemic damage, which can activate fibroblasts in a paracrine manner. Collectively, our data indicate phase-specific changes in cellular heterogeneity during different stages of cardiac remodeling and allow for the identification of therapeutic targets relevant for cardiac repair.

[1] Hubrecht Institute, Royal Netherlands Academy of Arts and Sciences (KNAW) and University Medical Centre, Utrecht, The Netherlands.
[2] Department of Cardiology, University Medical Centre, Utrecht, The Netherlands. [3]These authors contributed equally: Bas Molenaar, Louk T. Timmer.
✉email: e.vanrooij@hubrecht.eu

Heart failure due to ischemic heart disease is a worldwide health problem[1,2]. Although reperfusion therapy to relieve the arterial occlusion has significantly reduced the mortality rate in patients, long-term survival and quality of life remain poor due to progression into heart failure[3]. To effectively combat the lethality caused by heart failure, a more complete understanding of the wound healing process after ischemic injury is required.

Ischemia-reperfusion (IR) injury activates a plethora of cellular and molecular processes for wound healing in the heart, which are crucial determinants for clinical outcome[4–7]. In a coordinated process involving fibroblasts, neutrophils, macrophages, lymphocytes and endothelial cells, lost cardiomyocytes in the oxygen-deprived area are replaced by scar tissue, which can lead to impaired function and, subsequently, heart failure[4,5,8,9]. Changing the function or recruitment of these cells in the infarcted heart affects cardiac function and mortality in pre-clinical studies[9–16]. Although there is an increasing understanding of processes within and between cell types that are relevant for cardiac repair, many mechanisms and factors are poorly understood or are yet to be identified[4–7].

Historically, high-throughput, unbiased gene expression studies of the infarcted heart were limited in resolution as they had to be performed on cardiac tissue or on populations of cells isolated by fluorescence-activated cell sorting (FACS) based on expression of a gene marker. These types of studies failed to determine cell-specific gene expression, thereby preventing examination of transcription heterogeneity within cell populations or intercellular communication between cell types within the same tissue. Whole-transcriptome analysis at single-cell resolution overcomes the previous technical limitation, providing a more detailed view on gene expression changes occurring during various cardiac pathologies. Recently, we performed single-cell RNA sequencing (scRNA-seq) on the injured mammalian adult heart, allowing us to study gene expression profiles in all main cardiac cell types[17,18].

Here, we used scRNA-seq to get an in-depth view of the cellular and molecular changes within and between cells during multiple phases of the wound healing response following IR. Our data corroborated previously determined functions of multiple cell types during the different repair processes, but additionally revealed the regulation of currently unstudied factors. Based on known ligand-receptor pairing, we also used these data to establish potential intercellular communication networks between all main cardiac cell types during each phase of the wound healing response. As such, we found elevated expression of multiple unstudied secreted factors by stressed cardiomyocytes following IR injury, for which we could also detect an increase in circulating levels. Subsequent in vitro experiments suggested that one of these factors, beta-2 microglobulin (B2M), is able to stimulate the wound healing response via the activation of fibroblasts.

Together, our dataset provides detailed information on gene expression differences occurring within and between different cell populations during several stages of cardiac remodeling after ischemic damage. Using these data, we can now link gene expression alterations to changes in cellular function and use them to define new cellular interactions relevant for repair. Ultimately, a full understanding of all these cellular communications and molecular processes could provide a basis for novel therapeutic interventions to improve the wound healing response following ischemic injury of the heart and prevent progression into heart failure.

## Results

### Single-cell sequencing reveals dynamic cell type composition during repair of ischemic injury.
Healing of the heart after ischemic injury is a dynamic process involving multiple cell types. Upon ischemic damage, the distribution and function of different cell types in the damaged myocardium changes over time (Fig. 1a)[4,5]. This organized process suggests strict regulation and coordination of cell recruitment and function during all phases of the wound-healing response. To impartially study how this regulation is achieved, we performed scRNA-seq of cardiac cells at different phases of the healing response after IR in mice.

To do so, we applied a FACS-based protocol, which was previously shown to capture various cardiac cells from both healthy and ischemic hearts for downstream sequencing, including cardiomyocytes, fibroblasts, endothelial cells and macrophages[18]. This protocol uses a gating strategy based on various scattering properties and DAPI negativity to obtain viable cells for downstream sequencing. As cardiomyocytes are large and fragile cells, we aimed to ascertain our sorted cells were viable and intact. To this end, we re-incubated our sorted (living) cells with DAPI and observed that a high proportion of our cardiac cells remained viable (94% DAPI negative) throughout the whole procedure of tissue digestion followed by FACS (Fig. 1b and Supplementary Fig. 1a, b). Although DAPI negativity is used to mark viable cells with good membrane integrity by preventing DAPI from entering the cell, it could also indicate the absence of a nucleus in the selected cell fragment (e.g. by capturing non-nucleated cellular debris). To confirm that DAPI-negative cells captured by our FACS-based protocol were nucleated, we co-stained cells with DAPI and DRAQ5, a nuclear dye that stains both living and dead cells[19]. Our data indicated that 98.3% of all DAPI-negative cardiac cells captured with our gating strategy were positive for DRAQ5 (Fig. 1c), confirming the far majority of the cells to be viable (DAPI negative) and to contain a nucleus (DRAQ5 positive).

Next, we collected cells from hearts 1, 3 and 14 days post IR injury (1, 3 and 14 dp IR) and 1 and 14 days post sham surgery (1 dp Sham and 14 dp Sham). For each condition we collected cells from at least three different mice and used them for downstream scRNA-seq using the SORT-seq protocol[20]. After filtering for quality (see methods), a total of 2201 cells were used for analysis. The k-medoids clustering of $1 -$ pearson's $r$ identified a total of 11 robust clusters (Fig. 1d, e, and Supplementary Fig. 1c). Analyses of the gene expression profile of each cluster (Supplementary Data 1) revealed most cell types known to be present during the healing response after IR, including cardiomyocytes, fibroblasts, endothelial cells, macrophages and neutrophils (Fig. 1f). We next looked whether any clusters were enriched in cells obtained from a specific condition (Fig. 1g). In line with the expectations, we found a clear enrichment of neutrophils coming from 1 dp IR hearts (cluster 11)[4,5]. In addition, macrophages were divided into two clusters, one enriched in cells obtained 1 dp IR (cluster 9) and one enriched in cells obtained from 3 dp IR hearts (cluster 8). Although we also noticed an increase in the proportion of fibroblasts coming from 1 dp IR and 3 dp IR hearts, this increase was not as apparent as with neutrophils and macrophages (Fig. 1g).

Based on these data we concluded our data to be of high quality and reliable, thereby providing us the opportunity to study cellular and molecular changes within and between cells during multiple phases of the wound healing response following IR.

### Ischemic injury induces a hypertrophy-associated gene program in a subset of cardiomyocytes.
The biological function of various cell types changes over time during the ischemic wound healing response (Fig. 1a), but many factors involved in the temporal regulation of cell function are yet to be identified. To find potential new genes regulating cell type function over time,

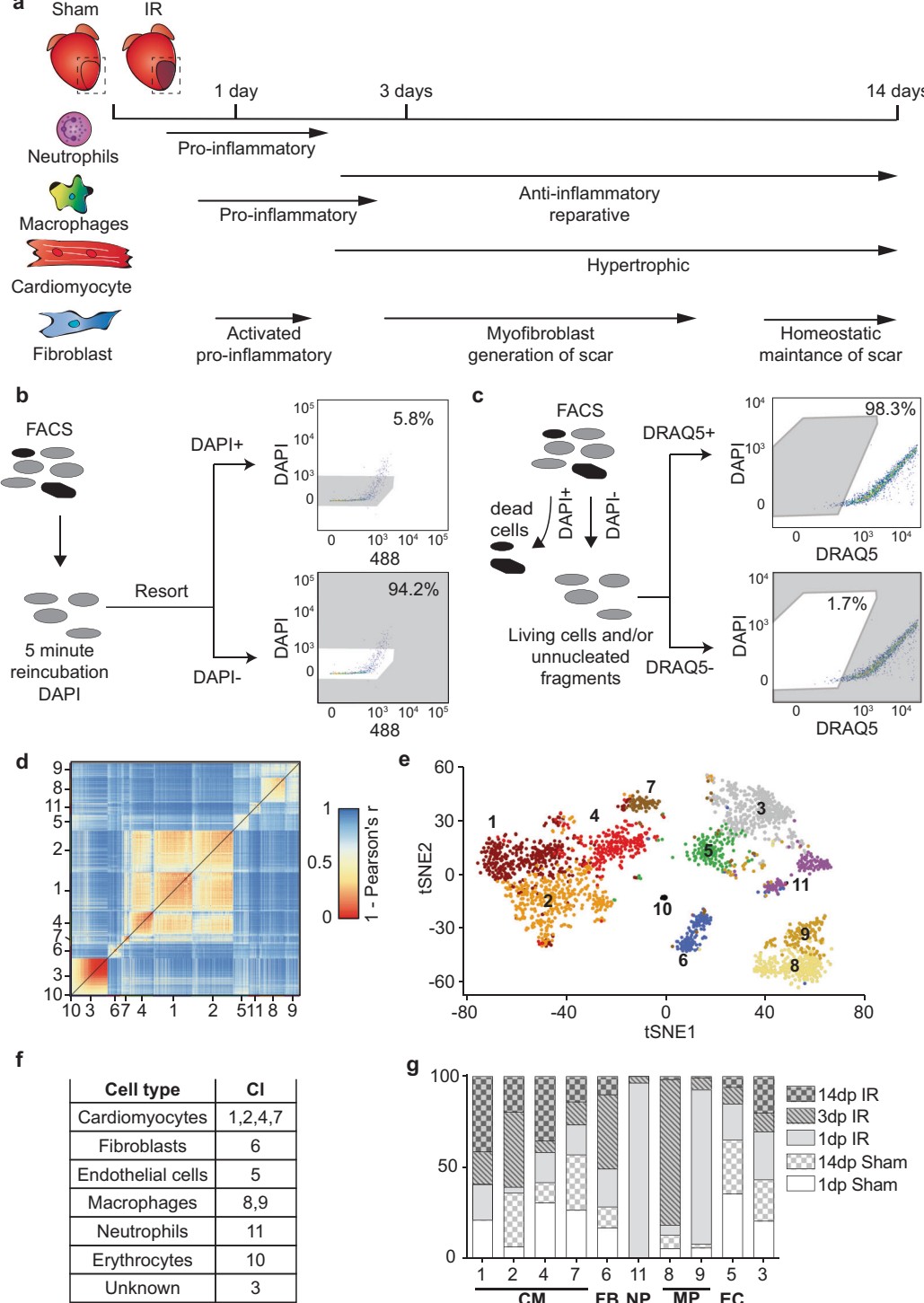

**Fig. 1 Single-cell sequencing reveals changes in the cellular composition of the myocardium at different time points after ischemic injury. a** Schematic of the study outline indicating time points of tissue harvest after ischemia/reperfusion (IR) injury and representation of the known cellular composition and function at these time points. **b** Determination of cell viability by DAPI negativity following dissociation and FACS procedure. **c** Schematic of the experimental setup to determine the percentage cells that are DAPI− but are nucleated as determined by DRAQ5 positivity. **d** Heatmap of the cell-to-cell transcriptome similarities (1 − Pearson's correlation coefficient) of 2201 cells obtained from all conditions combined. Cells are clustered based on transcriptome similarity using k-medoids clustering. **e** t-Distributed Stochastic Neighbor Embedding (tSNE) plot indicating transcriptome similarities across individual cells. Different colors and numbers highlight the clusters identified by k-medoids clustering in (**d**). **f** Table highlighting cell types that were characterized by the clustering analysis. **g** Bar graph showing the proportion of cells originating from the different condition per cluster. CM cardiomyocytes, FB fibroblasts, MP macrophages, NP neutrophils, EC endothelial cells.

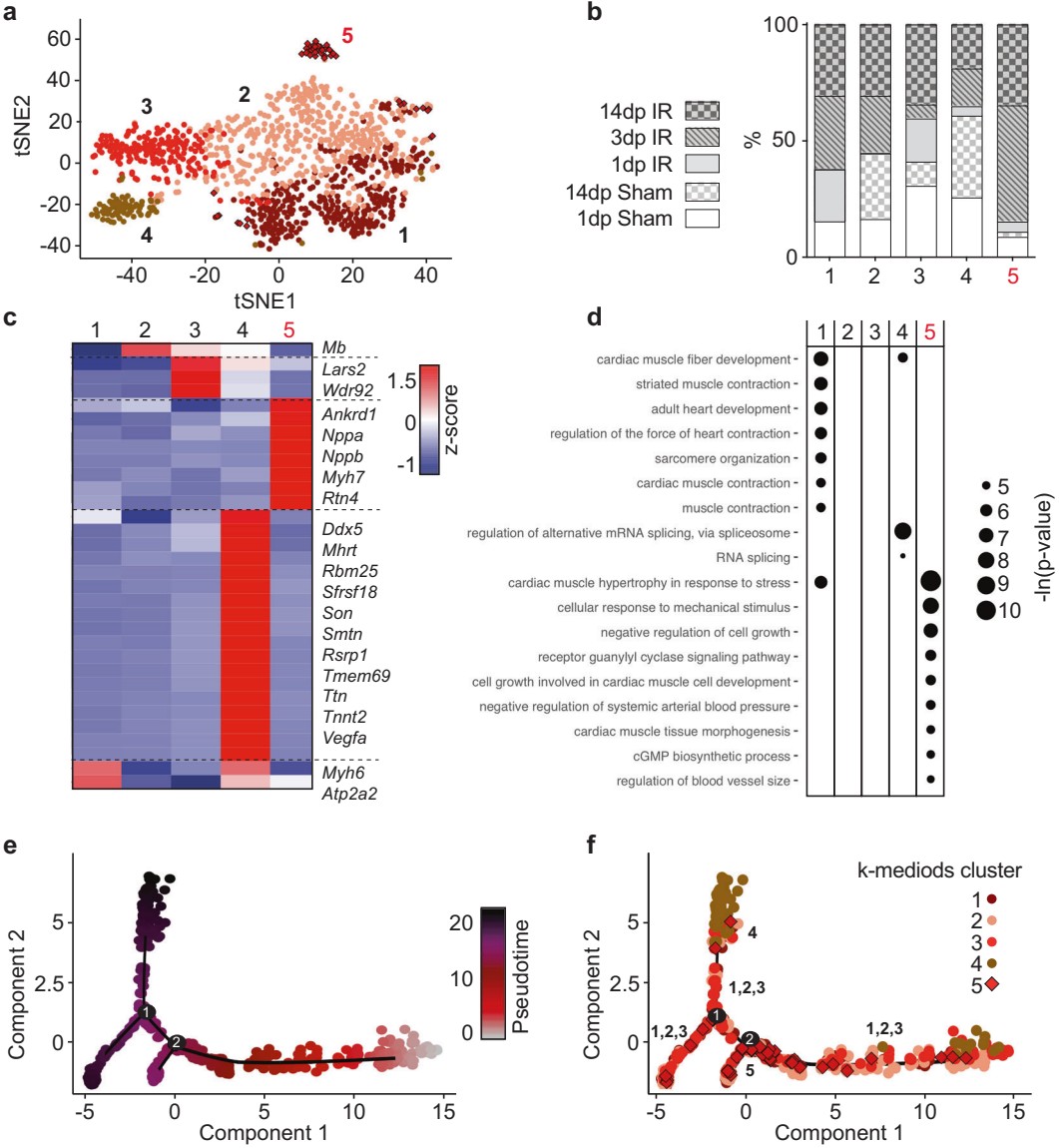

**Fig. 2 Single-cell sequencing indicates cardiomyocyte heterogeneity. a** tSNE plot indicating transcriptomic similarities across cardiomyocytes obtained from all conditions. Different colors, symbols and numbers highlight the clusters as determined by k-medoids clustering of 1 − Pearson's correlation coefficient. **b** Bar graph showing the proportion of cardiomyocytes originating from the different condition per cluster. **c** Heatmap showing expression of all genes significantly enriched in at least one cluster. Examples of genes enriched per cluster are depicted on the right side of the heatmap. **d** Bubble plot of top GO terms in gene ontology analysis on genes significantly enriched within the cluster. **b–d** Numbers highlighted in red depict a cluster that mainly contains 3 dp IR and 14 dp IR cardiomyocytes (cluster 5). **e, f** Cell trajectory analysis of cardiomyocytes, showing pseudotime on a color-coded scale (**e**) or highlighting the k-medoids cluster of each cardiomyocyte in the trajectory plot (**f**). The regions in the branches of the trajectory plot that contain mostly cells from k-medoids clusters 1,2,3 or cluster 4 or cluster 5 are highlighted by number.

we explored transcriptomic dynamicity within various cell types during different phases after IR.

To do so, we first selected all cardiomyocytes identified in all conditions, followed by refined subclustering of this cell type. Our clustering strategy revealed five cardiomyocyte subclusters with a mediocre transcriptomic similarity between each other (Fig. 2a and Supplementary Fig. 2a—intercluster distance of 0.57 ± 0.080 (mean ± SD)). Of these, clusters 1–4 contained cardiomyocytes from both sham conditions, as well as from different time points post IR (Fig. 2b). Clusters 1–3 did not clearly separate in two-dimensional space with tSNE. Further analysis of cluster-specific gene expression profiles revealed that these clusters show transcriptomic heterogeneity of well-established cardiomyocyte markers such as myosin heavy chain 6 (*Myh6*) and ATPase

sarcoplasmic/endoplasmic reticulum Ca2+ transporting 2 (*Atp2a2*) (Fig. 2c, Supplementary Fig. 2b–d and Supplementary Data 2). Expression of these markers occurs in continuous gradients throughout the whole cardiomyocyte cell population, rather than being limited to a discrete set of cells. Cluster 4 did form a distinct cluster and was characterized by a unique set of genes that were only dispersedly expressed across cardiomyocytes in other clusters, including vascular endothelial growth factor A (*Vegfa*), SON DNA binding protein (*Son*) and DEAD-box helicase 5 (*Ddx5*) (Fig. 2c, Supplementary Fig. 2e and Supplementary Data 2).

While all cardiomyocyte clusters contained cells from all conditions, cluster 5 appeared enriched for cells coming from hearts collected 3 and 14 dp IR (Fig. 2b). Analysis of the gene

profile revealed highly expressed genes classically associated with cardiac stress, including myosin heavy chain 7 (*Myh7*), natriuretic peptide A (*Nppa*) and natriuretic peptide B (*Nppb*) (Fig. 2c, Supplementary Fig. 2f and Supplementary Data 2)[21,22]. Indeed, this cluster showed gene ontologies for cardiac hypertrophy, cell growth and response to mechanical stress (Fig. 2d). In addition to known markers for cardiac stress, we identified various genes previously not clearly linked to hypertrophic cardiomyocytes, including Reticulon 4 (*Rtn4*) (Supplementary Fig. 2f). *Rtn4* was previously only shown to be dysregulated in ischemic cardiomyopathy on whole tissue level but, to the best of our knowledge, was never shown to be specifically upregulated in a subset of cardiomyocytes with a hypertrophy-associated transcriptome[23,24].

To further confirm transcriptomic heterogeneity in the cardiomyocytes, we additionally performed pseudotime and cell trajectory analysis on all cardiomyocytes using Monocle2[25] (Fig. 2e, f). Similar as in the tSNE plot, where clusters 1–3 did not form distinct clusters in tSNE space, cells from these clusters also did not form distinct branches but instead dispersed over multiple branches and interspersed with each (Fig. 2a, e, f). The branches containing most of the cells of clusters 1–3 showed comparable gradients of cardiomyocyte-marker expression as observed in the tSNE plot (Supplementary Fig. 2b–d). In contrast to clusters 1–3, cells from cluster 4 were predominantly present in one branch (upper branch—Fig. 2f). Cell from cluster 5, the cluster with a hypertrophic transcriptional profile, were mainly aggregated on one location in the trajectory plot (around branch point 2—Fig. 2f). Expression of genes enriched in cluster 5 was also highly expressed around this branch point (Supplementary Fig. 2f).

Taken together, our dataset indicates transcriptomic heterogeneity between cardiomyocytes, resulting in different subpopulations in the stressed and unstressed heart. Identifying a hypertrophy-associated gene expression profile enables us to define and study the functional relevance of genes that were previously unknown for their role in cardiomyocyte hypertrophy.

**Transcriptomic changes associated with cellular function of macrophages and fibroblasts.** Macrophages and fibroblasts are known to have multiphasic functions during wound healing post ischemia. Both move from a pro-inflammatory state, characterized by the release of pro-inflammatory cytokines and metalloproteases, to a pro-wound-healing state marked by the release of anti-inflammatory factors (Fig. 1a)[4,5,8]. During the wound-healing phase, fibroblasts also secrete pro-angiogenic factors and extracellular matrix (ECM) proteins to enhance scar formation[5,8]. To identify the factors involved in driving the cellular fate of fibroblast and macrophages in response to injury, we set out to determine relevant gene expression changes in these specific cells.

Subclustering of macrophages identified in all conditions formed eight clusters with a lower transcriptomic similarity between all clusters compared to cardiomyocyte clusters (Supplementary Fig. 3a—intercluster distance of 0.64 ± 0.09 (mean ± SD)), but these clusters were not clearly separated in tSNE space (Fig. 3a). Similar to cardiomyocytes, heterogeneity in transcriptomes between cell clusters was mainly driven by continuous gradients of gene expression (Supplementary Fig. 3b–d), which was previously also found in macrophages from healthy hearts[26]. Although clusters were not clearly separated, they were enriched in either 1 dp IR macrophages (clusters 2, 3, 5 and 8) or 3 dp IR macrophages (clusters 4, 6 and 7) (Fig. 3b). We also observed a cluster containing a substantial number of macrophages originating from sham hearts and from hearts during the intermediate and chronic phases of wound healing (cluster 1) (Fig. 3b). Genes

that were previously found enriched in resident cardiac macrophages were also higher expressed in this cluster (Fig. 3c, Supplementary Fig. 4 and Supplementary Data 3)[27]. We therefore concluded that cells from cluster 1 most likely constitute the resident macrophages.

Clusters enriched in 1 dp IR macrophages were placed closer to each other in tSNE space compared to other clusters (Fig. 3a, b), which was also the case for the clusters enriched in 3 dp IR macrophages. This indicates that the heterogeneity within the 1 dp IR and 3 dp IR macrophage populations prevents clear separation of these populations at a single-cell level. However, at the whole population level, the average transcriptome of the 1 dp IR macrophages does differ from that of 3 dp IR macrophages.

When looking at the gene expression profile of 1 dp IR macrophages we saw a high expression of various pro-inflammatory cytokines/chemokines associated with M1 macrophages. These included interleukin 1β (*Il1β*), chemokine (C-C motif) ligand 2 and 9 (*Ccl2* and *Ccl9*) and chemokine (C-X-C motif) ligand 3 (*Cxcl3*) (Fig. 3c, Supplementary Fig. 3c and Supplementary Data 3). Gene ontologies for these clusters were also associated with various pro-inflammatory processes (Fig. 3d and Supplementary Fig. 5). Surprisingly, Arginase 1 (*Arg1*) expression was high in pro-inflammatory 1 dp IR macrophages, whereas this gene is an established marker for anti-inflammatory M2 macrophages. However, this finding is in line with other single-cell sequencing datasets of the heart showing a comparable effect[27,28].

Macrophages of clusters enriched in 3 dp IR cells expressed genes more associated with the anti-inflammatory M2 macrophage polarization, including apolipoprotein E (*Apoe*), galectin 3 (*Lgals3*) and transmembrane glycoprotein NMB (*Gpnmb*) (Fig. 3c, Supplementary Fig. 3d and Supplementary Data 3). In addition to genes known to be relevant for macrophage biology, we also identified differentially expressed genes, like *Itm2b*, which have, to our knowledge, currently not been studied in cardiac macrophages before (Fig. 3c, Supplementary Fig. 3d and Supplementary Data 3).

Pseudotime and cell trajectory analysis on all macrophages confirmed the average transcriptomic differences in macrophages on population level between different time points after IR[25] (Fig. 3e, f). Macrophages from cluster 1 and clusters enriched in 1 dp IR or 3 dp IR macrophages separated predominantly in different branches (upper branch for cluster 1, lower right branch for 1 dp IR and lower left branch for 3 dp IR). In addition, expression of genes characterizing 1 dp IR or 3 dp IR macrophage clusters was also enriched in corresponding branches (Supplementary Fig. 3b–d). We were able to confirm the differential expression of *Arg1* and *Gpnmb* on whole tissue by qPCR (Fig. 3g).

Together, these results emphasize a gradual shift in macrophage transcriptomics from a pro-inflammatory M1 state in the acute phase to a pro-wound-healing M2 state in the intermediate phase on the whole cell population level.

In contrast to cardiomyocytes and macrophages, fibroblasts actually did separate in four distinct clusters with low transcriptomic similarity (Fig. 4a and Supplementary Fig. 6a—intercluster distance of 0.75 ± 0.05 (mean ± SD)). Two clusters (clusters 1 and 2) contained cells from all conditions and had a mediocre expression of ECM proteins (Fig. 4b, Supplementary Fig. 6b, d and Supplementary Data 4), probably representing quiescent resident fibroblasts. Cluster 4 was highly enriched in fibroblasts obtained 1 dp IR, whereas cluster 3 predominantly contained cells originating from 3 dp IR hearts (Fig. 4b). The 1 dp IR cluster expressed various pro-inflammatory chemokines and metalloproteases and showed genes ontologies associated with cell proliferation, translation and inflammation (Fig. 4c, d, Supplementary Fig. 6c and Supplementary Data 4). The 3 dp IR cluster

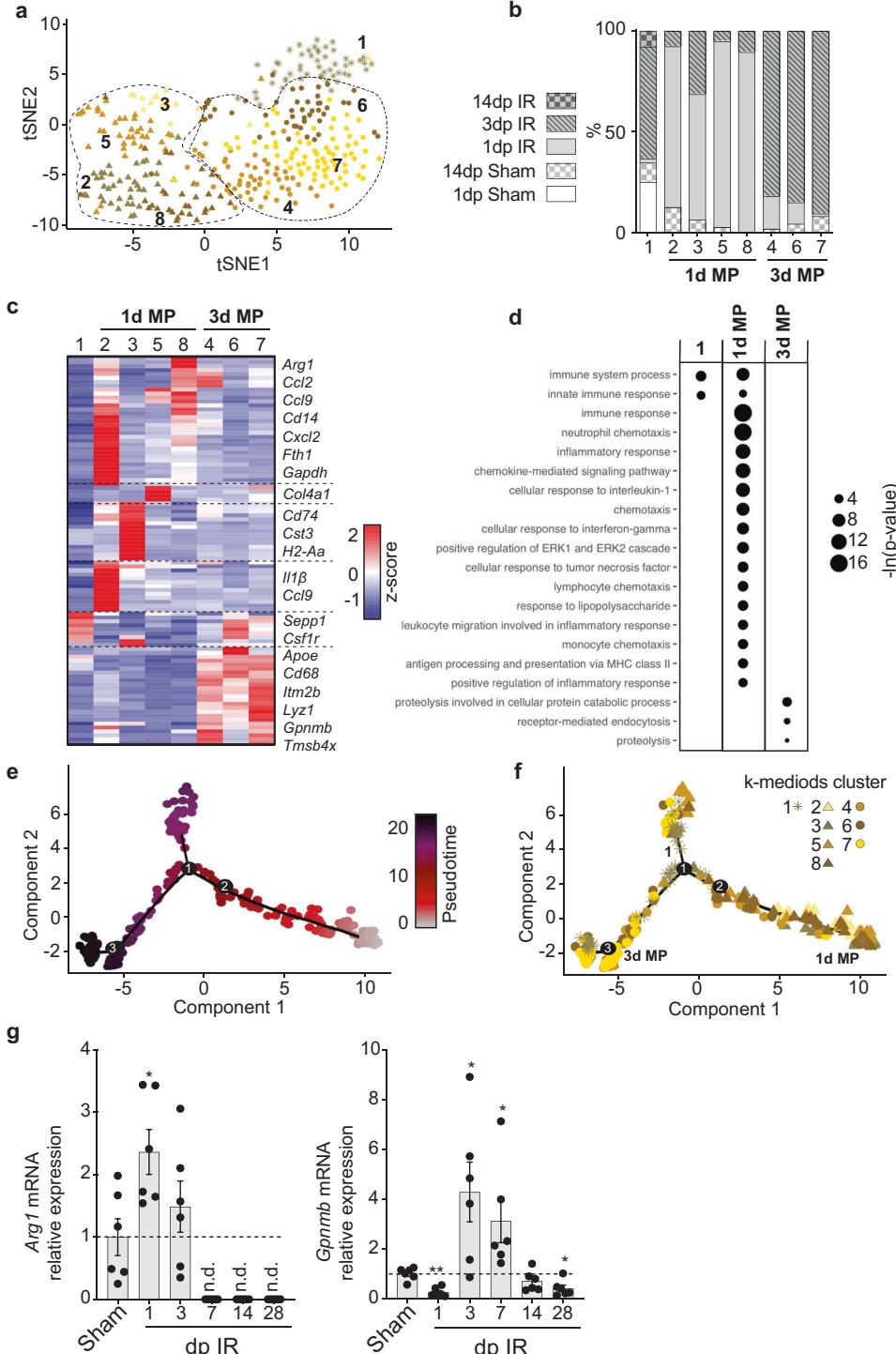

**Fig. 3 Macrophages show transcriptional heterogeneity after ischemia and shift from a pro-inflammatory to a wound healing transcriptional profile.**
**a** tSNE plot indicating transcriptomic similarities across macrophages obtained from all conditions. Different colors, symbols and numbers highlight the clusters as determined by k-medoids clustering of $1-$ Pearson's correlation coefficient. Stars highlight a cell cluster formed by macrophages from all conditions; triangles highlight clusters containing mainly 1 dp IR macrophages and circles depict clusters containing mainly 3 dp IR macrophages. **b** Bar graph showing the proportion of macrophages originating from the different conditions per cluster. **c** Heatmap showing expression of all genes significantly enriched in at least one cluster. Examples of genes enriched per cluster are depicted on the right side of the heatmap. **d** Bubble plot of top GO terms in gene ontology analysis on genes significantly enriched within either cluster one, in at least one of the 1 dp IR enriched clusters or in at least one of the 3 dp IR clusters. **e, f** Cell trajectory analysis of macrophages, showing pseudotime on a color-coded scale (**e**) or highlighting the k-medoids cluster of each macrophage in the trajectory plot (**f**). The branches in the trajectory plot that contain mostly cells from cluster 1, from the 1 dp IR clusters or from the 3 dp IR clusters are highlighted by number or tekst. **g** RT-qPCR analysis on the infarcted cardiac tissue at different time points post IR for *Arg1* and *Gpnmb*. Per time point post IR, expression levels are relative to the corresponding time point post sham surgery ($n = 6$ animals). Data are presented as mean ± SEM. *$P < 0.05$, **$P < 0.01$, Two-sample *t*-test vs corresponding time point post sham, with Holm–Sidak adjustment for multiple comparisons. n.d. not determined.

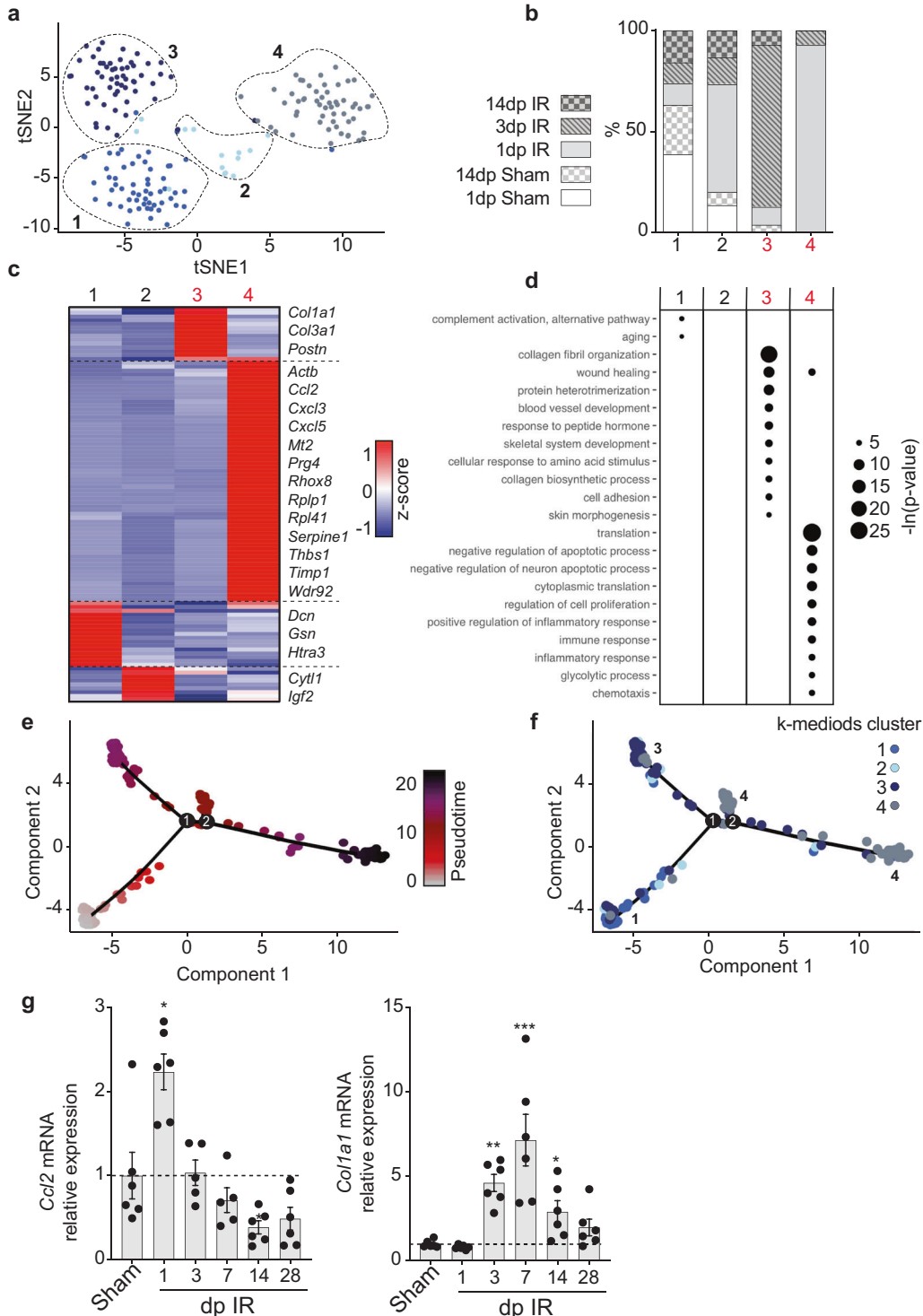

**Fig. 4 Transcriptome-wide changes in fibroblasts at different stages after ischemic injury suggest dynamic fibroblasts function during wound healing.**
**a** tSNE plot indicating transcriptomic similarities across fibroblasts obtained from all conditions. Different colors and numbers highlight the clusters as determined by k-medoids clustering of 1 − Pearson's correlation coefficient. **b** Bar graph showing the proportion of fibroblasts originating from the different conditions per cluster. **c** Heatmap showing expression of all genes significantly enriched in at least one cluster. Examples of genes enriched per cluster are depicted on the right side of the heatmap. **d** Bubble plot of gene ontology analysis on genes significantly enriched within a cluster. **b**–**d** Numbers highlighted in red depict clusters that mainly contain 1 dp IR (cluster 4) or 3 dp IR (cluster 3) fibroblasts. **e**, **f**, Cell trajectory analysis of fibroblasts, showing pseudotime on a color-coded scale (**e**) or highlighting the k-medoids cluster of each fibroblast in the trajectory plot (**f**). The branches in the trajectory plot that contain mostly cells from k-medoids cluster 1, 3 or 4 are highlighted by number. **g** RT-qPCR analysis on the infarcted cardiac tissue at different time points post IR for Ccl2 and Col1a1. Per time point post IR, expression levels are relative to the corresponding time point post sham surgery (n = 6 animals). Data are presented as mean ± SEM. *P < 0.05, **P < 0.01, ***P < 0.001, Two-sample t-test vs the corresponding time point post sham, with Holm–Sidak adjustment for multiple comparisons.

had high expression of ECM proteins and showed gene ontologies for wound healing, ECM deposition and neovascularization (Fig. 4c, d, Supplementary Fig. 6d and Supplementary Data 4). Again, our analysis enabled the identification of differentially expressed genes between different wound healing phases that, to our knowledge, have not been related to fibroblast function before (e.g. *Wdr92*). Single-cell trajectory analysis and qPCR confirmed the time-dependent differences in fibroblasts' transcriptomes, going from an inflammatory state to a scar formation state (Fig. 4e–g). As our data lacked sufficient fibroblasts obtained from the late time point post IR (14 dp IR), we were unable to gain information on fibroblast function during the chronic phase.

Taken together, our analyses show the dynamics of cell function over time in response to ischemic injury. Consistent with current literature, both macrophages and fibroblasts show an inflammatory function during the acute phase, followed by a shift towards a pro-wound healing function in the intermediate phase after ischemia[4,5]. Future studies into the function of the newly identified genes showing an enrichment during the different stages of remodeling could provide valuable information about the intracellular mechanisms that regulate cell function dynamics during the wound healing response.

**Single-cell transcriptomics identifies a communication network between cardiomyocytes and other cardiac cells relevant after ischemic injury.** So far, other studies identifying intercellular communication in healthy and ischemic mouse hearts have lacked cardiomyocytes in their datasets due to the cellular size restrictions of their approaches[26,27]. A compelling advantage of our FACS-based technology is the opportunity to additionally collect data from cardiomyocytes.

Using our data we were able to define intercellular communication networks of all main cardiac cells at different time points after ischemic injury using a dataset of ligand-receptor pairs (Fig. 5a, b, Supplementary Fig. 7a–f and Supplementary Data 5-9)[26,29]. Focusing on the immediate phase after ischemic injury (1 dp IR), cell types found in the infarct region included cardiomyocytes, fibroblasts, macrophages, neutrophils and endothelial cells (Fig. 5a). Analysis of ligand-receptor expression between different cell types revealed an extensive network of potential intercellular communication by paracrine signaling between cell types (Fig. 5b and Supplementary Data 9), providing a possible system by which these cell types can coordinate their function. Close examination of the individual cell types revealed fibroblasts to express most ligands, for which recipient cell types expressed their cognate receptor (Fig. 5c, Supplementary Fig. 7b, d, f and Supplementary Data 9).

Next, we aimed to investigate how stressed cardiomyocytes 1 day after ischemic injury could potentially influence other cell types by paracrine signaling. For this, we first screened for ligands expressed in cardiomyocytes for which a substantial proportion of the other cell types express the cognate receptor (Fig. 5d). We found that *Nppa* and *Nppb*, two well-established factors secreted by cardiomyocytes, were the only factors specifically secreted by cardiomyocytes[30]. To increase the likelihood that screened ligands play an important role during the acute phase post ischemia, we selected only ligands that are upregulated threefold or more in cardiomyocytes 1 dp IR compared to 1 dp sham. This yielded 149 ligand-receptor combinations, which included 39 unique ligands. Next, we discarded ligands that are part of the ECM or for which the expression was highly enriched in another cell type (e.g. *Il1β*). This resulted in a set of 15 ligands that are expressed by and upregulated in cardiomyocytes 1 day after ischemia, for which one or more cognate receptors are expressed by another cell type (Supplementary Fig. 8).

From these 15 we selected three factors for further analysis: milk fat globule-EGF factor 8 protein (*Mfge8*), calreticulin (*Calr*) and *B2m* (Fig. 5e). We were able to detect these factors in the blood plasma of mice both under healthy conditions and detected an increase in circulating levels during the immediate and intermediate phases of wound healing following IR (Fig. 6a, b and Supplementary Fig. 9). In addition, human cardiomyocytes obtained from iPS cells also express and secrete these factors (Supplementary Fig. 10a–c).

As fibroblasts expressed the highest level of the cognate receptors of *Mfge8*, *Calr* and *B2m* (Fig. 5e and Fig. 6c), we next supplemented fibroblast cultures with the recombinant protein of MFGE8, CALR or B2M to determine the functional effect of this interaction. Of the three selected factors, B2M induced the expression of the myofibroblast markers actin alpha 2, smooth muscle (*Acta2*) and *Vimentin (Vim)* (Fig. 6d). In addition, B2M stimulated fibroblast migration in a scratch assay (Fig. 6e, f), an in vitro model for wound healing. Immunohistochemistry indicated that B2M is induced after IR in mice and predominantly expressed in the infarcted area by both cardiomyocytes as well as other infiltrating cells (Fig. 6g).

These results reveal the potential intercellular communication networks between all main cardiac cell types during each phase of infarct healing and suggest that increased secretion of B2M by stressed cardiomyocytes post IR, contributes to the activation of fibroblasts. These results extend current knowledge about potential paracrine signaling of stressed cardiomyocytes, a crucial process in the repair process after ischemia[31,32].

## Discussion

Although we are starting to understand the function of and crosstalk between cardiac cell types in response to injury, to date many processes remain poorly understood[4–7]. In this study, we used a FACS-based scRNA-seq method on the adult, injured heart to detail the dynamics in cellular distribution, function and communication at different time points after ischemic damage. In doing so, we were able to collect genome-wide transcriptomic data for all main cardiac cell types providing us with detailed information on gene expression differences occurring within and between different cell populations during several stages after ischemic damage. Based on these data we were able to connect changes in gene expression to alterations in cellular function and use them to define new cellular interactions relevant for the heart. This is exemplified by our discovery that cardiomyocytes express and secrete elevated levels of B2M in response to ischemic damage, which can activate fibroblasts in a paracrine manner. Using our data to improve our knowledge on cardiac repair and remodeling could eventually contribute to the development of novel therapeutic interventions for patients suffering from ischemic heart disease.

Initial analysis of our sc-RNA-seq data underscored the validity of our approach by indicating the expected dynamics in cell-type distribution and cell function throughout the wound healing response[4,5,33]. Neutrophils were only detected during the acute phase after ischemic injury, while the population of macrophages and fibroblasts expanded during the acute and intermediate phase after injury followed by a regression during the chronic phase. Functionally, based on gene expression changes, the macrophages and fibroblasts switched towards an anti-inflammatory and pro-wound healing/pro-angiogenic state, respectively. Similar switches have previously also been found in bulk-RNA sequencing data of isolated macrophages and fibroblasts[28,34]. However, unlike what was previously proposed, our data showed a continuum in the transcriptional profile of macrophages from a pro-inflammatory (M1) towards an anti-inflammatory (M2) state, rather than the

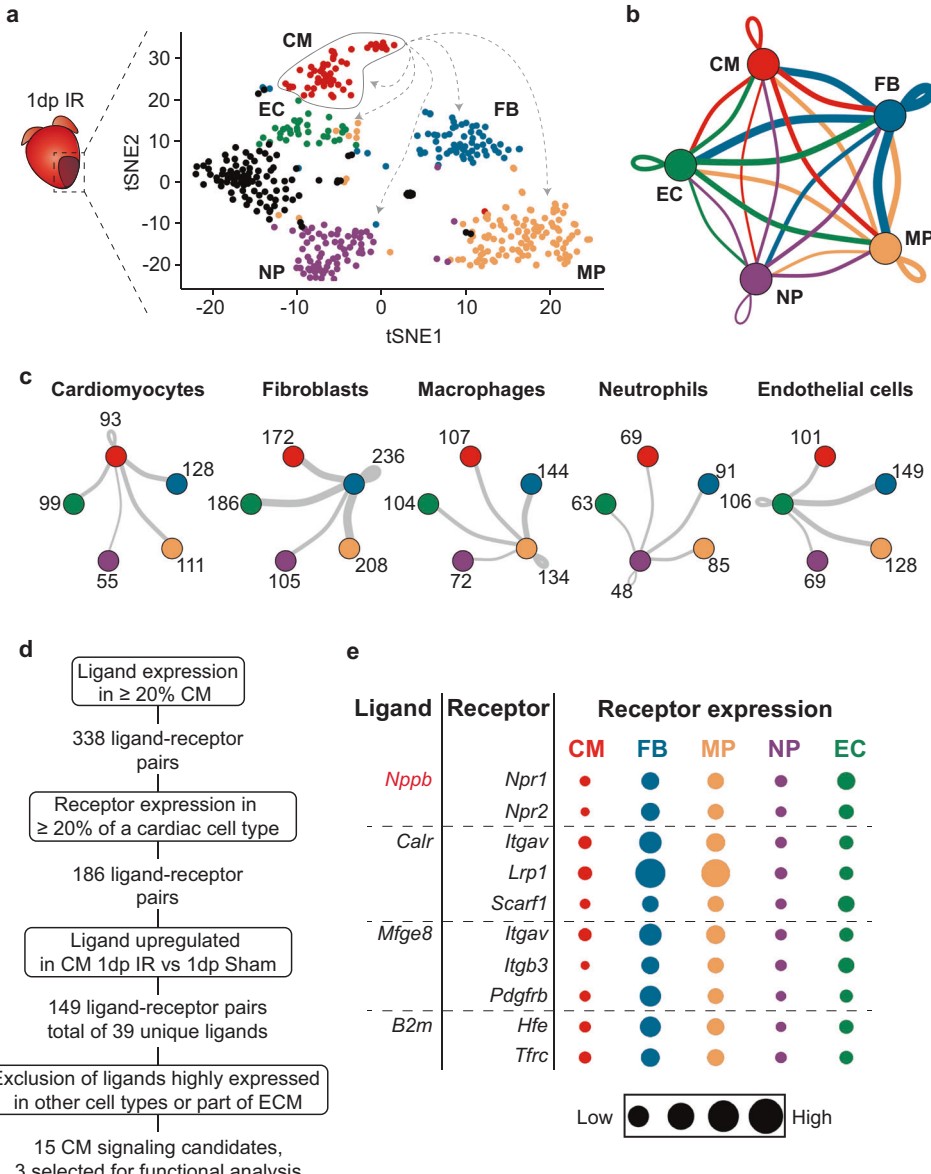

**Fig. 5 Intercellular communication by autocrine and paracrine signaling in the ischemic heart. a** tSNE plot of all cells obtained 1 dp IR. Colors indicate different cell types. CM cardiomyocytes, FB fibroblasts, MP macrophages, NP neutrophils, EC endothelial cells. To determine potential autocrine/paracrine signaling after ischemia, expression of ligands in one cell type was determined (e.g. cardiomyocytes highlighted by circle). Subsequently, the expression of cognate receptors was determined per cell type (highlighted by dashed arrows). **b** Spider graph illustrating the potential intercellular communications between cell types by ligand-receptor signaling. The line color depicts ligand expression in the cell type with the same color. Lines connect to cell types that express the corresponding receptor. Line thickness is proportional to the number of ligands expressed in one population for which the receptor is expressed in the other cell type, with loop indicating autocrine signaling. **c** Potential intercellular ligand-receptor signaling between cell types, described for each cell type separately. Numbers depict the number of ligand-receptor couples between inter-cell type links. **d** Schematic of the screening strategy used to identify potential ligands involved in autocrine/paracrine signaling between stressed cardiomyocytes and other cell types 1 dp IR. **e** Bubble plot showing ligands selected for downstream functional study with their cognate receptor(s). A positive control (Nppb) is depicted in red. The size of the bubble represents the expression of each receptor across all identified cardiac cell types 1 dp IR.

presence of two distinct subsets of macrophages[27,28,32,35]. This suggests that the conventional M1/M2 polarization might be less distinct than what was previously suggested.

Although macrophages and fibroblasts are known to be present and have distinct functions during later stages of wound healing[8,34], the small number of these cells captured 14 dp IR prevented us from reliably studying flux and cell function at this phase. This is probably due to a limitation of our single-cell sequencing protocol, which was designed to preferentially collect larger cells[17,18]. Future studies capturing a greater number of

non-cardiomyocytes together with cardiomyocytes will enable a more detailed analysis of how these cell types behave during the chronic repair phase.

Among the sequenced cardiomyocytes we identified a distinct subpopulation of cells characterized by a transcriptomic profile that is highly associated with stress-related hypertrophy. The observation that only a subset of cardiomyocytes show a hypertrophy-associated transcriptional profile after IR is consistent with findings in a previous study where IR was shown to induce hypertrophy specifically in cardiomyocytes flanking the

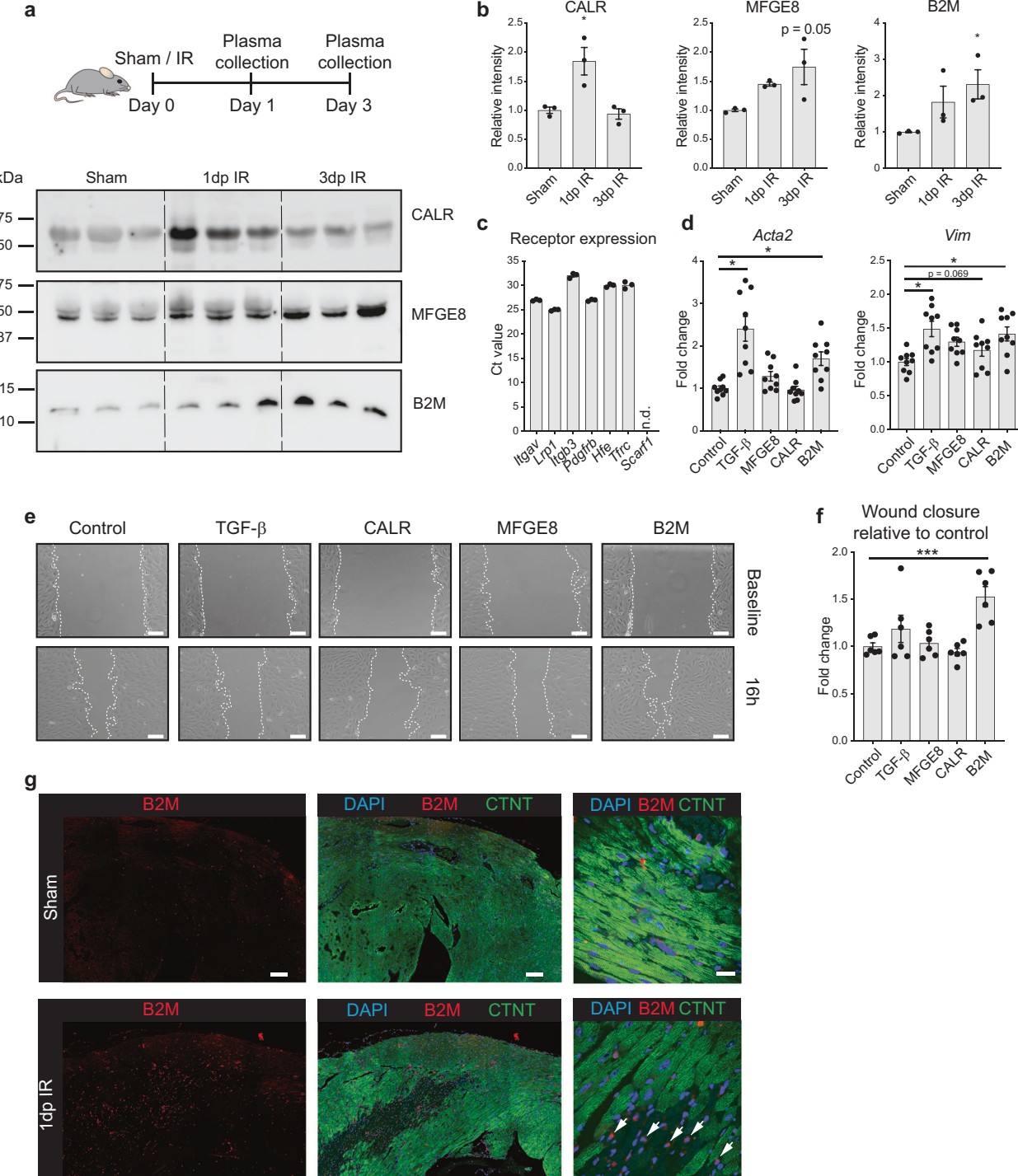

**Fig. 6 B2M is secreted by stressed cardiomyocytes to activate fibroblasts. a** Western blot on mice plasma at different time points post IR or 1 day post sham (*n* = 3 animals). Equal loading of plasma proteins was confirmed by Coomassie staining (Supplementary Fig. 9). **b** Quantification of immunoblots in (**a**); *P < 0.05 compared to sham, Kruskal–Wallis test with Dunnett's multiple comparison post-hoc test. **c** Cognate receptor expression of selected ligands in NIH/3T3 fibroblasts is shown by RT-qPCR. **d** Upregulation of two myofibroblast (*Acta2* and *Vim*) markers in NIH/3T3 cells after treatment with B2M, TGF-β served as positive control (*n* = 9 from 3 independent experiments). **e** Increased wound healing of NIH/3T3 cells after treatment with B2M, assessed by scratch assay. Images show representative pictures for each condition. Dashed lines indicate gap size. **f** Quantification of wound healing assay relative to corresponding control (*n* = 6 from 2 independent experiments). Scale bars, 200 μm. **g** Immunohistochemistry images of heart slices after IR or sham surgery. White arrows indicate cardiomyocytes (larger cells with diminished CTNT staining) within the infarct zone that show B2M expression. CTNT cardiac troponin T. Scale bars, 100 μm (left and middle panels) and 20 μm (right panels). Data are presented as mean ± SEM. *P < 0.05, ***P < 0.001. One-way ANOVA with multiple comparison vs control and Dunnett's adjustment for multiple hypothesis testing.

fibrotic regions[16]. Although the relative number of viable cells, indicated by DAPI−, does not significantly differ between the different conditions and timepoints (Supplementary Fig. 1b), we cannot rule out that the proportion of hypertrophic cardiomyocytes is underrepresented because of increased digestion difficulty of fibrotic areas, where these cells are located (Fig. 2a, cluster 5). Nonetheless, in addition to known marker genes for cardiomyocyte hypertrophy these cells do reveal an enrichment for genes that were previously unrecognized for a role in this process. Functional follow-up studies are needed to define their functional relevance during cardiomyocyte hypertrophy.

With our single-cell sequencing dataset we were also able to generate an unbiased intercellular communication network between all main cardiac cell types after IR based on a previously generated draft network of ligand-receptor pairs[29]. Similar to results obtained in a previous study in healthy hearts, we observed fibroblasts to be the most communicative cell type during all phases[26]. They express the most ligands for which other cell types express the cognate receptor. The communication networks generated here should help to study how all main cardiac cell types can communicate with one another over the time course of the wound healing response following ischemic injury.

From the 15 identified paracrine factors released from cardiomyocytes, we functionally pursued three factors: B2M, MFGE8 and CALR. Of these, B2M was found to stimulate a wound healing response in fibroblasts. *B2m* is widely used as a housekeeping gene, but its use has been shown to be inappropriate in myocardial infarction studies[36]. *B2m* is a component of the major histocompatibility complex (MHC) class I molecules and present on the cell surface of basically all nucleated cells. Based on its immunogenic role as a component of the MHC class I molecule, *B2m* has been deleted in umbilical mesenchymal stem cells (UMSC) with the rationale of improving stem cell engraftment after MI[37]. Treatment with *B2m*-deficient UMSC was superior in attenuating fibrosis and improving cardiac function post MI compared to wild-type UMSC treatment. A potential role of *B2m* as a signaling factor remained, however, uninvestigated. Our results suggest that secreted B2M is involved in the fibrotic response during the myocardial repair process after ischemia, which could contribute to the improved wound healing observed in rats treated with either *B2m* deficient UMSC or their exosomes[37].

The inability of MFGE8 and CALR to induce a clear effect on fibroblasts might indicate that the suggested ligand-receptor interaction is not relevant in this particular setting in vitro, but does not necessarily exclude a role in vivo as the interactions are likely cell type and context dependent. Additionally, it should be noted that the identified ligand-receptor interactions are often not unique between two specific cell types. Different cell types can often co-act to enable ligand-receptor interactions. This also appears to be the case for B2M as B2M was also expressed in other cardiac cell types than cardiomyocytes (Supplementary Fig. 11a–e and Supplementary Data 9). Based on our data we know that increased B2M expression from cardiomyocytes contributes to fibroblast activation during the acute phase post IR, however, this does not exclude other cell types to stimulate fibroblasts via B2M as well.

Since the secretion of B2M and MFGE8 progressively increased up to 3 dp IR, we analyzed which ligands would still be selected 3 dp IR. We observed that the number of selected ligands 3 dp IR was much smaller compared to 1 dp IR (Supplementary Fig. 12), suggesting a less pronounced role in intercellular communication during the intermediate phase of wound healing. Our data is consistent with transcript level changes preceding changes in ligand secretion into the serum. We reason that changes in transcript levels precede changes in protein secretion levels due to the general delay in protein synthesis/decay, which cause a temporal delay between induction/repression of transcription of a gene and corresponding changes at the protein level[38].

While several other studies have performed single-cell sequencing analysis on healthy and diseased cardiac tissue[27,35,39–45], one key advantage of our FACS-based technology is the fact that our approach allows for the collection and sequencing of intact adult cardiomyocytes. This is unlike the approaches used in other studies, where they were restricted by cell size or focused on sequencing nuclei only.

A limitation of our approach is the inherent bias towards isolating larger cells defined by the gating parameters of the flow sorter[17,18]. As a consequence, the relative abundance of cardiomyocytes in our dataset is higher than one would expect based on the cell-type composition of the heart (47% of cells in our dataset are cardiomyocytes, whereas approximately 30% of cells in the heart are cardiomyocytes)[46]. As such, our dataset is less suitable to study the relative abundance of cell types in the heart during one timepoint. However, since the same inherent bias towards larger cells is present during all timepoints, relative changes in the abundance of a cell type between the different timepoints can be studied.

In conclusion, our single-cell sequencing study provides an extensive, unbiased dataset of numerous processes occurring in each cell type during different stages of the repair process upon IR injury. In addition to confirming known processes and factors, we identified potential new players that could have essential functions during specific repair phases in the different cell types. We also generated a database of potential intercellular-communication networks between all main cardiac cell types for each phase of the wound healing response. We believe that these results can serve as a resource for the development of novel insights and a deeper understanding of the repair process and intercellular crosstalk after ischemic injury of the heart. A complete detailed understanding of all these cellular and molecular interactions will result in multiple potential therapeutic strategies to improve the wound healing process after ischemic cardiac injury.

## Methods

**Experimental animals and IR surgery.** All animal studies were performed in accordance with institutional guidelines and with approval of the Animal Welfare Committee of the Royal Netherlands Academy of Arts and Sciences. All animals used were of C57BL/6 J background and 8 weeks old.

Ischemia-reperfusion was performed by a 1-h temporal ligation of the left anterior descending (LAD) artery. The whole surgical procedure and following recovery period occurred at 38 °C. Mice were anaesthetized with intraperitoneal injections with a mixture containing ketamine and xylazine. Hairs on the skin anterior of the thorax were shaved. A tracheal tube was placed, and the mouse was connected to a ventilator (UNO Microventilator UMV-03, Uno BV). The surgical site was cleaned using iodine and 70% ethanol. In aseptic conditions and with sterilized surgical instruments, an incision was made at the midline, allowing access to the left third intercostal space. Pectoral muscles were retracted, and a caudal incision was made to expose the third rib. Next, wound hooks were placed to allow access to the heart. The pericardium was incised longitudinally and the LAD was identified. To achieve temporal ischemia, LAD was ligated by placing a A7.0 silk suture around the LAD together with a 2–3 mm PE 10 tubing. After 1 h of ischemic injury, reperfusion of the myocardium was achieved by cutting the ligature. After surgery, mice were injected with 0.5–0.1 mg/kg of buprenorphine (temgesic). Next, wounds were closed with a 5.0 silk suture and skin was closed with wound clips. The animal was disconnected by removing the tracheal tube and placing under a nose cone with 100% oxygen. For sham surgery, a similar procedure was performed with the exclusion of a 1-h ligation of the LAD artery. Hearts were collected 1 day after the sham surgery (1 dp Sham), or 1 day (1 dp IR) and 14 days (14 dp IR) after the IR surgery. For the time points 14 days after sham (14 dp Sham) and 3 days after IR (3 dp IR) surgery, datasets were used that were previously published[18]. For plasma collection, blood was withdrawn from the left ventricle into an EDTA blood tube (Greiner bio-one, #450475). These tubes were centrifuged at 2000×*g* at 4 °C for 15 min to capture plasma.

**Cardiac tissue dissociation into single-cell suspension.** Dissociation of cardiac tissue into single-cell suspension was done according to an extended protocol

described previously[18]. In short, mice were euthanized and the heart was exposed. Before removal, the heart was perfused by removing the right atrium and injecting 10 ml ice-cold perfusion buffer consisting of 135 mM NaCl, 4 mM KCl, 1 mM MgCl$_2$, 10 mM HEPES, 0.33 mM NaH$_2$PO$_4$, 10 mM glucose, 10 mM 2,3-butane-dione monoxime, 5 mM taurine; pH 7.2. Afterwards, the infarcted area of the left ventricle was removed and carefully minced on ice in perfusion buffer and next placed in digestion buffer consisting of 0.5 mg/ml Liberase TL (Roche, #540102001), 20 µg/ml DNase1 (Worthington, LK003172) and 10 mM HEPES, dissolved in 1.305 ml DMEM (Dulbecco's Modified Eagle Medium, high glucose, GlutaMAX Supplement, pyruvate (Gibco, #31966021)). Digestion occurred at 37 °C in a shaking water bath for 15 min. Subsequently, the cell suspension was carefully pipetted up and down and passed through a 100 µm cell strainer (EASYstrainer, #542000, Greiner Bio-One). The remaining tissue pieces were gently rubbed through the strainer using a plunger of a 1 ml syringe (#303172, BD Plastipak). Next, the strainer was rinsed with 8.5 ml of DMEM and centrifuged at 300 × g for 6 min at 4 °C. The supernatant was removed and the pellet containing cells was carefully resuspended in 2 ml ice-cold DMEM with 3 µM DAPI (Invitrogen, #D3571) and immediately used for downstream single-cell sorting.

**Flow cytometry and cell viability/RNA quality control**. First, 5000 cells were sorted from the cell suspension into a 96-well plate using FASCAria II (BD bioscience). These cells were imaged using EVPS Cell Imaging Systems for checking cell viability by visualization of the morphology. Single cells from the cell suspension were sorted into a 384-well plate. Cells were selected using the gating strategy as described before[18]. In short, selected cells were DAPI negative, had a high autofluorescence at 488 and 460 nm, and had an elongated morphology as determined by the FSC-Height and FSC-Width.

To determine the RNA quality for each sort, an additional of 100, 500, 1000, 5000 and 10,000 cells were sorted into a 96-well plate containing TRIzol (Fisher Scientific). RNA extraction was performed according to the manufacturer's instructions. Afterwards, RNA quality was determined by Bioanalyzer (Agilent 2100). 384-well plates containing single cells were only used for downstream single-cell sequencing if the RIN of these cells in bulk was above 7.5.

**Determination of membrane integrity and nuclear presence in sorted cells**. To show that our gating strategy selects cells with good membrane integrity, we applied our standard strategy as described above and captured cells in five different FACS tubes. These cells were again stained for DAPI and after 5, 15, 30, 45 or 60 min incubation, DAPI negativity was again determined by flow cytometry. The selection of cardiac cells based on DAPI negativity secures the collection of viable cells during FACS, but can also be explained by a lack of a nucleus. To confirm that all DAPI-negative cells have a nucleus, additional DRAQ5 staining (d1:1000, Thermofisher Scientific, #65-0880-92) was performed before sorting. DAPI only stains nuclei of dead cells, whereas DRAQ5 stains nuclei of both live and dead cells[19]. Dissociated cardiac cells were sorted with the same gating strategy used to obtain single cells for sequencing. Thereafter, nuclear presence in these cells was determined by FACS by excitation of DRAQ5 at 638 nm.

**Library preparation and single-cell sequencing**. After sorting single cells, cDNA preparation was performed using the SORT-seq protocol[20]. Cells were sorted into 384-well plates containing 5 µL VaporLock oil and an aqueous solution of 100 nL containing reverse transcriptase (RT) primers, spike-in RNA molecules, dinucleotide triphosphates (dNTPs) and CEL-seq primers. CEL-seq primers consisted of a 24 bp polyT sequence followed by a 6 bp unique molecular identifier (UMI), a cell-specific barcode, the 5′ Illumina TruSeq2 adapter and a T7 promoter sequence. Cell lysis was formed by incubation of cells for 5 min at 65 °C, after which cDNA libraries were obtained by dispersion of the RT enzyme and second strand mixes with the Nanodrop II liquid handling platform (GC biotech). cDNA libraries in all wells were pooled and, after separation of the aqueous phase from the oil phase, in vitro transcribed for linear amplification by overnight incubation at 37 °C. Next, Illumina sequencing libraries were prepared using the TruSeq small RNA primers (Illumina) and sequenced paired-end at 75 bp read length with Illumina NextSeq.

**Preprocessing of single-cell sequencing data**. Paired-end reads from Illumina sequencing were mapped with BWA-ALN to the reference genome GRCm38/mm10 downloaded from the UCSC genome browser[47]. The left read mate was used to allocate reads to the libraries and cells, whereas the right read mate was mapped to the gene models and used to quantify transcript abundance. Reads that mapped equally well to multiple loci in the reference genome were excluded. For quantification of transcript abundance, per cell-specific barcode the number of transcripts containing unique UMIs was counted for each gene in the gene model. Next, these observed transcript counts were converted into expected transcript counts using Poissonian statistics as previously described[48], taking into account the count number for each gene and a total of 4096 different UMIs. Afterwards, all read counts for mitochondrial genes were discarded due to the high abundance of transcripts coming from these genes in cardiomyocytes, which interfered with downstream clustering. In addition, reads mapped to Rn45s were removed since transcripts of this gene do not have a poly-a tail, and detection of this gene is a known artefact of the SORT-seq procedure (not published). Per condition, we

sequenced between 3–8 plates coming from 3–4 mice. The biological replicates were independently obtained from separate litters.

**Analysis of single-cell sequencing data and clustering strategy**. Single-cell RNA sequencing data were analyzed using RaceID2 in R version 3.5.2.. Expected transcripts counts were used to cluster cells based on k-medoids clustering, to visualize cell clusters using t-distributed stochastic neighbor embedding (tSNE) and to compute genes significantly up- or downregulated in all cells in the cluster compared to cells not in the cluster[49]. Per cell, total read counts were downsampled to 1000 unique transcripts, and cells that had a number lower than 1000 unique transcripts were discarded. For k-medoid clustering and tSNE plotting of 1 − pearson's $r$, all genes were used that had a downsampled read count of at least 3 in at least one cell (min.expr = 3, min.number = 1, max. expr = inf). Outlier detection in the RaceID2 algorithm was switched off. For robust k-medoids clustering the following strategy was used: First, clustering was performed over the data using the default cluster parameters, where the number of k-medoid clusters is determined by the RaceID2 algorithm based on saturation of within-cluster dispersion. Next, cluster robustness was determined by calculating Jaccard's coefficient over all bootstrapped samples. Clusters were considered robust if at least all clusters except one had a Jaccard coefficient > 0.6. This yielded robust clusters for the full dataset, all cells obtained from 1 dp sham and 14 dp sham hearts combined, and all macrophages across all conditions. If clustering did not achieve this level of robustness, iterative k-medoids clustering was performed by reducing the number of clusters in steps of one until the level of robustness was achieved. This achieved sufficient clustering robustness in the subset of the data containing all cells from 1 dp IR, 3 dp IR and 14 dp IR, and in the subsets containing all cardiomyocytes and all fibroblasts across all conditions.

**Differential gene expression and gene ontology**. To determine enriched genes expressed in a cluster, the function *clustdiffgenes* was used in the RaceID2 algorithm, which determines the differential expression of genes between cells in a specific cluster and all cells not in the cluster. A *p*-value is subsequently calculated based on binomial counting statistics. The profile of enriched genes of a cluster was used to identify the cell type described by the cluster. To generate heatmaps, the average normalized expression of all genes significantly enriched in at least one cluster was selected and transformed into z-scores. Next, gene ontology (GO) analysis was performed on these genes using DAVID[50]. GO-terms were filtered using a false discovery rate (FDR) filter of 10% to account for multiple comparisons. Bubble plots show the −ln transformed *p*-values of Fisher's exact test. Bubble plots were generated using ggplot2 for R[51].

**Quantitative real-time PCR**. Total RNA was isolated from the infarct zone of heart ventricles with TRIzol reagent (Life Technologies) according to the manufacturer's instructions. Total RNA (1 µg) was applied to mRNA based reverse transcription using an iScript cDNA Synthesis Kit (Bio-Rad). Real-time PCR was performed according to based SYBRgreen methodology (Bio-Rad). Transcript quantities were normalized for endogenous loading using *Hprt or Gapdh*. The used primer sequences are shown in Table 1. The thermocycling parameters were as following: denaturation at 95 °C for 15 s, annealing at 60 °C for 30 s and extension at 72 °C for 30 s.

**Cell trajectory and pseudotime analysis**. The cellular trajectory across expression space and pseudotime was also determined using Monocle2[25] on subsets of data containing one cell type across all conditions (macrophages and fibroblasts). First, all cells belonging to the same cell type across all conditions were selected and used as input. To determine the cell trajectory over time, the dpFeature protocol was used for correct input. For analysis, only genes expressed in at least 5% of all cells were used. Next, tSNE dimensionality reduction was performed on the first two or three principal components for all fibroblasts or macrophages, respectively. The number of principal components used for tSNE dimensionality reduction was determined by the amount of variability explained by each principal component in the subset of data. Next, clusters were determined using the density peak clustering on default settings, and differential expression of genes across all cells was determined. For cell trajectory and pseudotime analysis, the top 1000 significantly differentially expressed genes were used.

**Prediction of intercellular communication**. Potential intercellular communication networks were determined using previously compiled ligand-receptor pairs[29]. Per cell type, the percentage of cells expressing the ligand or cognate receptors by a read count of at least 1 was determined (Supplementary Data 5). Next, ligands and receptors were regarded as "expressed" by a cell type if the expression was found in at least 20% of cells[29]. Next, potential communication links between two cell types were defined by determining the expression of the ligand in the first cell type and expression of the receptor in the second cell type. Networks were plotted using igraph for R[52]. For the selection of potential ligands upregulated by cardiomyocytes after IR, differential expression for genes was calculated between all cardiomyocytes obtained from 1 dp sham and 1 dp IR using the DESeq2 package for R[26,53].

**Table 1 qPCR primers used.**

| Gene | Sequences |
| --- | --- |
| *Arg1* - mouse | Fw 5'- ctcaaaaggacagcctcgagga, Rv 5'- cccgtggtctctcacgtcatac |
| *Gpnmb* - mouse | Fw 5'- gaatgggatgaacacctgtatcc, Rv 5'- ccacaaaagtgatattggaaccc |
| *Ccl2* - mouse | Fw 5'- ggctcagccagatgcagtt, Rv 5'- tctccagcctactcattggga |
| *Col1a1* - mouse | Fw 5'- aatgcaatgaagaactggactg, Rv 5'- ccctcgactcctacatcttctg |
| *Acta2* - mouse | Fw 5'- ttctataacgagcttcgtgtgg, Rv 5'- gagtccagcacaataccagttg |
| *Vim* - mouse | Fw 5'- atcagctcaccaacgacaag, Rv 5'- aatgactgcagggtgctttc |
| *Hprt* - mouse | Fw 5'- agcctaagatgagcgcaagt, RV 5'-atggccacaggactagaaca |
| *Gapdh* - mouse | Fw 5'- tgtcgtggagtctactggtg, Rv 5'- acaccatcacaaacatgg |
| *Itgav* - mouse | Fw 5'- acgtcctccaggatgtttctcct, Rv 5'- ttgccctccttctacaatccca |
| *Lrp1* - mouse | Fw 5'- cgagctctgtgaccagtgtt, Rv 5'- aagagcacacgatgccttca |
| *Itgb3* - mouse | Fw 5'- gacaactctgggccgctct, Rv 5'- cacgcctcgtgtggtacag |
| *Pdgfrb* - mouse | Fw 5'- tgcagagacctcaaaaggtg, Rv 5'- cctgatcttcctcccagaaa |
| *Hfe* - mouse | Fw 5'- ccaccgcgttcacattctct, Rv 5'- ccacatagcccctagcctca |
| *Tfrc* - mouse | Fw 5'- ctcgtgaggctggatctc, Rv 5'- tagcatggaccagtttacc |
| *Scarf1* - mouse | Fw 5'- gctggtcagcacgtctgtaa Rv 5'- cacagatggggatggtgcat |

The left column contains the official gene symbol followed by the species it was used to detect. The right column contains the sequences of the forward (Fw) and reverse (Rv) primer from the 5' to 3' end.

**Cell culture.** Human iPS cells were obtained from the LUMC hIPS Core facility (LUMC0099iCTRL04) and cultured in Essential 8 Medium (Thermofisher Scientific, A1517001) on Geltrex-coated plates (Thermofisher Scientific, A1413302). For differentiation into cardiomyocytes, a previously described protocol was used[54]. In short, cells were cultured until 80–90% confluency. On day 0, the medium was replaced with differentiation medium: RPMI medium (Thermofisher Scientific, 72400021) containing 0.5 mg/ml recombinant human albumin (Sigma, A9731) and 0.2 mg/ml L-ascorbic acid (Sigma-Aldrich, A8960). The differentiation medium was supplemented with 5 µM CHIR99021 (Sigma, SML1046). At day 2, the medium was refreshed with a new differentiation medium supplemented with 5 µM IWP2 (Millipore, 681671). At days 4 and 6 the medium was again refreshed with a new differentiation medium. On day 8, cultures were transferred to cardio-culture medium: RPMI medium (Thermofisher Scientific, 72400021) containing B-27 supplement (Thermofisher Scientific, 17504001). The medium was refreshed every two days for a total of two times. Afterwards, the iPS-cell cultures were exposed to glucose-depleted/lactate-supplemented medium to select for cardiomyocytes. This was achieved by transferring the cells to a selection medium: RPMI 1640 without glucose without HEPES (Biological Industries, 01-101-1 A) containing 0.5 mg/ml recombinant human albumin (Sigma, A9731), 0.2 mg/ml L-ascorbic acid (Sigma-Aldrich, A8960), 4 mM lactate (SC-301818A Chemcruz) and 3.5 mM HEPES (H0887 Sigma). Afterwards, cells were cultured in cardio-culture medium for approximately 2-3 days and allowed to recover. Next, the cells were dissociated with 10 X TrypLE™ Select Enzyme (Gibco, A1217701) and re-plated in geltrex-coated wells with or without coverslips in cardio-culture medium supplemented with 2 µM Thiazovivin (Millipore, 420220). The next day, the medium was refreshed with cardio-culture medium. After recovery for 2 days, the cells and supernatant were collected for downstream analysis.

NIH/3T3 fibroblasts were cultured in DMEM (Gibco, #31966-021) supplemented with 10% fetal bovine serum and 1% penicillin-streptomycin. NIH3T3 were treated with 100 ng/ml recombinant proteins of MFGE8 (R&D Systems, #2805-MF-050), B2M (Sino Biological, #50957-M08H) or CALR (Biomatik, #RPC24546). 5 ng/ml TGFB (Peprotech, #100-21) and equal amounts of BSA, which served as a positive and negative control, respectively. RNA expression was determined after 24 h of treatment. In vitro wound healing was assessed by a scratch assay. NIH/3T3 cells were plated in 24-well plates and scratches were made using a p200 pipet tip. At baseline and 16 h post-scratch images were taken, after which the gap closure was assessed relative to baseline using ImageJ.

**Histology and immunofluorescence.** Cells were fixed with 4% PFA, permeabilized with 0.1% Triton X-100 and blocked with 4% (for B2M and MGFE8) or 8% (for CALR and VEGF) goat serum. Subsequently, cells were incubated for 2.5 h at room temperature with primary antibodies against B2M (rabbit 1:50, Proteintech, 13511-1-AP) or MFGE8 (mouse 1:50, Santa Cruz, sc-271574), or incubated overnight at 4 °C with antibodies against CALR (mouse 1:50, Santa Cruz, sc-166837). Cells were co-stained using antibodies against Cardiac Troponin T (CTNT) (rabbit 1:200, Abcam, ab45932) or alpha-Actinin-2 (ACTN2) (mouse 1:800, Sigma, A7811). Next, cells were stained with corresponding Alexa Fluor antibodies (donkey anti-mouse IgG (H + L) 1:400, Invitrogen, A21202, or donkey anti-rabbit IgG (H + L) 1:400, Invitrogen, A10042) for 45 min at room temperature. Next, cells were washed and mounted before imaging. For staining of B2M expression and cardiomyocytes in cardiac tissue, whole hearts were fixed in 4% formalin for 48 h at room temperature and embedded in paraffin. Four-micrometer sections were cut and incubated in xylene and ethanol to water gradient for deparaffinization and rehydration. Antigen retrieval was performed by boiling the sections for 20 min in Tris-EDTA buffer (pH 9.0). After cooling down

for 30 min, sections were blocked in 0.5% BSA in PBS for 30 min at room temperature. After blocking, sections were incubated with antibodies against B2M (rabbit 1:50, Proteintech, 13511-1-AP) and CTNT (mouse 1:250, Abcam, ab8295) in 0.5% BSA in PBS overnight at 4 °C. Next, a signal amplification kit (Alexa Fluor™ 594 Tyramide SuperBoost™ Kit, goat anti-rabbit IgG, Thermofisher, B40944) was used according to the manufacturer's instruction to amplify B2M staining. In short, sections were incubated for 60 min with a poly-HRP conjugated secondary antibody at room temperature, washed in PBS and subsequently incubated with Alexa Fluor™ 594 Tyramide reagent in reaction buffer for 10 min at room temperature. After applying a stop reagent, sections were washed in PBS and incubated with DAPI (1:1000, Invitrogen, #D3571) and a secondary antibody (Alexa Fluor™ 488 donkey anti-mouse IgG (H + L) d1:250, Invitrogen, #A21202) for 1 h at room temperature to visualize CTNT. Subsequently, sections were washed and mounted.

Immunohistochemistry of B2M (rabbit 1:50, Proteintech, 13511-1-AP) expression in combination with NIMP-R14 (rat 1:50, Abcam, ab2557), CD68 (rat 1:200, Bio-Rad, MCA1957), Vimentin (mouse 1:50, Santa Cruz, sc-373717) and CD31 (goat, 1:50, R&D Systems, AF3628) was performed on frozen tissue. Whole hearts were embedded in Tissue Freezing Medium (Leica Biosystems, # 14020108926) and stored at −80 °C. Ten-micrometer cryosections were cut using a cryostat and stored at −20 °C until use. Sections were thawed at RT and rehydrated in PBS. Next, sections were blocked in 0.5% BSA in PBS for 30 min at room temperature. After blocking, sections were incubated with primary antibodies overnight at 4 °C. Corresponding secondary antibodies (Alexa Fluor™ 488 donkey anti-mouse IgG (H + L) (d1:500, Invitrogen, #A21202), Alexa Fluor™ 488 donkey anti-goat IgG (H + L) (d1:500, Invitrogen, #A11055), Alexa Fluor™ 488 donkey anti-rat IgG (H + L) (d1:500, Invitrogen, #A21208) and Alexa Fluor™ 568 donkey anti-rabbit IgG (H + L) (d1:500, Invitrogen, # A10042) were applied for 1 h at RT together with DAPI (1:1000, Invitrogen, #D3571) before washing and mounting. Images were taken using a Leica TCS SPE confocal microscope and processed with Fiji software.

**Western blot.** For secreted protein detection by iPS-cell-derived cardiomyocytes, the cell culture supernatant was snap-frozen in liquid nitrogen and stored at −80 °C until further use. Protein concentration was quantified by Bradford assay (Bio-Rad). 25 µg (plasma) or 10 µg (supernatant) of protein was separated by SDS-PAGE in a 10% (MFGE8 and CALR) or 15% (B2M) acrylamide gel and analyzed by western blotting using antibodies against MFGE8 (1:250, % BSA in TBS-T Santa Cruz 271574), B2M (1:1000, 5% BSA in TBS-T, Proteintech #13511-1-1-AP) and CALR (1:250, 5% BSA in TBS-T, Santa Cruz Biotechnology #sc-166837). For plasma protein detection, HRP conjugated antibodies against CALR (1:100, 5% BSA in TBS-T, Santa Cruz Biotechnology #sc-166837 HRP) and MFGE8 (1:100, % BSA in TBS-T Santa Cruz Biotechnology #sc-271574 HRP) were used. Equal loading of plasma proteins was confirmed by Coomassie staining on a parallel gel. Blots were imaged using an ImageQuant Las4000 scanner (GE Healthcare Life Sciences).

**Statistics and reproducibility.** Statistical significance was assessed using unpaired two-tailed *t*-tests and one-way ANOVA's. Normality of data distribution was determined using the Shapiro–Wilk test. In case an ANOVA was performed, equality of variance was first tested using Brown–Forsythe test. If assumptions were not met, statistical tests were performed on log-transformed data or a Kruskal–Wallis test was applied (in case of the latter, this is stated in the figure legend). If required, *p*-values were corrected for multiple comparisons (used tests

are stated in the figure legends). Statistical details regarding the analysis of single-cell sequencing data are described in the 'Differential gene expression and gene ontology' section.

**Reporting summary**. Further information on research design is available in the Nature Research Reporting Summary linked to this article.

## Data availability
Raw data and processed read count tables are available online in the Gene Expression Omnibus (GEO) under accession code GSE146285. All source data for the figures are available as an excel file (Supplementary Data 10).

## Code availability
The source code for the RaceID2 algorithm can be accessed at https://github.com/dgrun/StemID. The code generated for the intercellular communication is available upon request from the Lead Contact.

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

## Acknowledgements

The authors thank Stefan van der Elst, Anko de Graaff and Jeroen Korving for technical assistance. This project has received funding from the European Union's Horizon 2020 research and innovation programme under the grant agreement No. 874764.

## Author contributions

B.M., L.T.T., M.D., M.M.G. and E.R. designed the experiments. L.T.T., B.M., M.D., J.M., H.R. and I.P. performed the experiments. B.M. analyzed single-cell data. D.V. and L.K. performed animal surgeries. B.M., LT.T., M.M.G. and E.R. wrote the manuscript.

## Competing interests

The authors declare no competing interests.
