## [Peer Review File · Communications Biology]

Reviewers' comments:

Reviewer #2 (Remarks to the Author):

I recommend that the paper be accepted for publication with minor edits and clarification to the manuscript. The project is well conceived, executed, and presented, and is thorough. It applies a relatively new technique (scRNA_seq) to study an important pathology, IR injury, and wound healing in general.

Minor questions/concerns:

- how many 384-well plates were processed per condition? I apologize if any of these information was already provided and I missed them.
- Were more than one mice pooled for each condition? Were there biological replicate from separate litter?
- Was there any corrections necessary for any batch effect? How were the datasets for different conditions integrated?
- Adult or terminally differentiated cardiomyocytes (CM) are often multi nucleation. Do the authors see any on facts? Were there any particular cluster that showed higher avg or median # of genes/transcripts that could be representative of these multi-nucleated CM?
- On a related note, it is very difficult to distinguish a whole CM from broken CM. I suppose the DRAQ5 vs DAPI staining is helpful here. what was the aspect ratio of these cardiomyocyte cells (i.e., length along long axis vs. short axis)?
- fig 1, clusters 6 (FB), 11 (NP) and 8,9 (MP) all seem like they have two clusters each. Broadly speaking, what are the clusters: IR vs sham? 1dp vs. 3dp? Or artifact from batch correction? (The individual cell-type clustering is too fine-grained to tell.) CM clusters 1,2,4,7 look fine in this respect. Could these cells have been used to integrate datasets?
- For the Hypertrophic CM, do % MT-genes show any trend across time-points?
- Are there any potential biases in cell-type recovery based on dissociation, e.g., 1dp vs 14dp IR? One would (perhaps naively) think that scar/fibrotic tissue may be harder to dissociate (in addition to the inherent challenges in dissociating cardiac tissue into single cell suspensions which is necessitating single nucleus RNA-seq in many studies)? What is cell viability % from each time-point? This may be calculated from the DAPI vs DRAQ5 fractions.
- In figure 3e, how should we interpret 1dp IR and 3dp IR on two separate trajectories?
- I really like the analyses, and the data are beautifully presented. Perhaps a bit more discussion to sum up the findings and highlight them in general terms? Some of the take-home messages may be made more general to make it more accessible to a wider community (I was overwhelmed reading the result and discussion sections, but the methods and figures were impressive. I particularly appreciated the ligand receptor analysis!)
- I don't quite see the utility of the iPSC-CM or fibroblast (FB) scratch experiments. It is not obvious from the optical images that the B2M treated FB heal better than the control. (Is 16 hr long enough to see this effect?) If there is one thing that the ligand receptor analysis shows, it is how ubiquitous communication among cell-types are. Perhaps a co-culture of CM and FB would have been more helpful and not just stimulating with certain factors/proteins. Even conditioned medium would be helpful. Anyway, the paper stands pretty well on its own without the in vitro experiments, and this section does not take anything away for it.

Reviewer #3 (Remarks to the Author):

The manuscript by Molenaar et al describe the use of single-cell RNA-sequencing to understand the role of cardiac cell gene expression changes at 3 key time points following ischemic injury. The authors identify the expression and secretion of beta-2 microglobulin by cardiomyocytes following ischemic injury, which in turn activates cardiac fibroblasts to promote scar formation. Overall, the manuscript is well written and organized. However, a few points would strengthen the overall impact of the paper. These include:

Major Comments

- 1) In the myocardium, while cardiomyocyte encompass the greatest volume of cells, they are lower in numbers relative to other cell types, most notably endothelial and fibroblast cells. The

data presented herein with sorting followed by scRNA-seq, suggest significant loss of both cell types. The authors should clarify how this may impact interpretation of the data.

2) Identification of subpopulations of cardiomyocytes in sham and ischemic myocardium is intriguing, but exploration into functional differences is limited (Fig. 2). The authors should include pseudotime plot analyses on cardiomyocytes, as shown in Figs. 3 & 4 to support their data.

3) Macrophages have been well-characterized to have both resident and infiltrating macrophage populations. Based on the current literature, how do your macrophage subpopulations (Fig. 3) compare to those that have been published? The authors should stratify their macrophage subpopulations to align with either a resident or infiltrating macrophage phenotype.

4) The expression of B2M increases at d3 relative to d1 post-MI (Fig. 6A). Does the expression of B2M (and its cognate receptors) change if you perform the same analyses in Fig.5 on 3d post-MI samples? Since these data are available, these analyses would strengthen the mechanism.

5) It is unclear how post-MI induced B2M expression, which is shown to increase in murine cardiomyocytes by scRNA-seq (Fig. 5) and cardiac tissue by WB (Fig. 6A), is localized to infiltrating immune cells by histology (Fig. 6A). Do macrophages play a significant role in B2M expression and modulation of fibroblast activity? This should be more clearly explored with additional supporting data.

6) All immunoblots in Fig. 6 should be quantitatively analyzed and included in the figure.

7) Histology in Fig. 6g should be analyzed with additional cell-specific antibodies to delineate the expression of B2M within each cell type described herein (i.e., cardiomyocytes, fibroblasts, macrophages, neutrophils, and endothelial cells), as alluded to in comment 5. Quantitative analyses of B2M expression in each cell type should be included at d1 and d3 post-MI.

Minor Comments

1) Fig. 1 e-g. You describe a cell population with unknown designation (3; Fig. 1e-f). How does population 3 shift at each time point, as you've shown with the other cell populations (Fig. 1g)?

2) Clarify heatmap key (Right part of pseudotime plots – Figs. 3 & 4), which reflects the colour-coding from tSNE plots (Figs. 3 & 4)

3) Since the iPS data (Fig. 6b) does not align with the murine model, it should be placed in the supplement.

4) Fig. 6e-f. The order and labeling in which these figures are placed is confusing. The quantitative measurements defining the change in area (Fig. 6f) should be placed after the representative images (Fig. 6e).

We thank the editor and the reviewers for their positive feedback and constructive suggestions to improve our manuscript. Please find below a point-by-point response to the issues that were raised in *Green Italic*

Reviewer 1

The work is meticulously done, and the assays are state of the art. The manuscript is well written and easy to follow. The novelty over previous work lies on the identification of new cell clusters and communications involved in the repair after IR and the identification of new potential biomarkers that can be used for deeper understanding of this process and could also significantly contribute to future therapeutic strategies.

We thank the reviewer for the positive feedback and suggestions for improvement, which we have tried to incorporate in this updated version of the manuscript.

- In figure 1f there is a misspelling mistake: Neutrophils instead of Neutrophiles
Thank you for pointing this out. We have now corrected this.
- In figure 1g: There is a progressive decrease on endothelial cells. Could you explain why do you think that is happening?
We think the relatively lower number of EC's are likely due to several potential reasons: 1) a higher abundance of other cell types, such as infiltrating immune cells and proliferating fibroblasts. 2) A necrosis of endothelial cells at the site of injury. 3) the transition of endothelial cells towards mesenchymal cells, which has been shown to occur post cardiac injury¹.
- In figure 2c: There are significant differences in cluster 3 (Lars2 and Wdr92) compared to the other clusters. Could you explain why that is happening and what could be the implication of those significant changes in this cluster of CM?
As cluster 3 is not associated with significant GO-terms we cannot reliably make conclusive statements regarding the function of this cluster of cardiomyocytes. However, based on the enriched expression of Wdr92 and Lars2 we can hypothesize about their function.
Lars2 codes for mitochondrial leucyl-tRNA synthetase and is enriched in tissues with high metabolic rates such as the heart and skeletal muscle tissue². As cluster 3 also is enriched in sarcomeric genes (Ttn and Actc1; Database S2) this might indicate a subset of cardiomyocytes with a higher metabolic demand due to enhanced sarcomeric content.
The Wdr92 loci has been associated with blood pressure³. Although we could not find relevant experimental data for this gene, we can imagine that the changes in blood pressure as a result of the reperfusion injury affects the expression Wdr92. In addition, the Wdr92 protein contains 2 WDR40 repeat domains, which have been implicated in TNF-alpha induced apoptosis⁴ and it could therefore also mark a subset of cells going into apoptosis (however, no other apoptotic markers are enriched in cluster 3).
However, functional follow up experiments will be required to make a definitive statement about the role of these proteins in cardiomyocyte biology.

- Results (line233-234): I could not find the results for *Itm2b* in Fig 3 or Supp Fig 3. Please clarify where these results can be shown.
We apologize for omitting these data in our original version and have now included it in Figure 3c and Supplemental Figure 3d.

Figure 3c

Supplementary Figure 3d

- Results (line 237-239): I find very curious the significant increase of Arg 1 at early stages, being a classical anti-inflammatory marker. I understand that this concept has been challenged recently, but how can you explain that Arg1 levels are too low to be detected when the anti-inflammatory response is happening (from 7 dp IR)?

While we agree with the reviewer that this result is unexpected, we think that several reasons could be causing the absence of detectable Arg1 levels at 7 dpIR. First of all, there is a decline in the amount of macrophages 7 days after ischemic injury compared to earlier stages⁵. Secondly, as pointed out by the reviewer, the role of Arg-1 as a classical M2 marker has been challenged recently^{5,6} and it was actually shown that <25% of classically activated bone marrow derived M2 macrophages express Arg-1⁶. This raises the option that the subset of macrophages expressing Arg-1 in the heart during the anti-inflammatory stage may be limited. Comparable results have previously been reported in other (single-cell) sequencing dataset of the heart after ischemic injury^{5,7}.

- Results (line 316): On which premises did you based the decision of studying these 3 genes (MFGE8, CALR and B2M). I could not understand why you chose those genes and not others.

As 15 candidate genes was too much to pursue we had to focus on a subset of ligands characterize for their downstream function. To do this, we picked candidates that had not been extensively characterized for their function in cardiomyocytes or cardiac cell-to-cell signaling. During this process we also excluded some candidates for practical reasons such as gene size, isoform differences or the lack of good antibodies.

It is important to highlight that while we decided to pick these 3, the other ligand candidates might also prove to be crucial when regulating other cell types. However, since we were unable to pursue all findings, we hope our dataset can be used as a useful resource for other groups that are interested in the other candidate ligands.

- Discussion: Is there any study that evaluate the contribution of MFGE8 and CALR that could explain, at least in part, why you do not see an effect on fibroblasts? Or maybe contradictory publications on the effect of these genes on fibroblast? If so, would be good to discuss about it in the discussion and not leave it only as a potential limitation of the in vitro set-up

We thank the reviewer for this suggestion. To strengthen our manuscript we have now reformulated our explanation for the absence of an effect of MFGE8 and CALR on fibroblasts. As the ligand-receptor interactions are putative interactions that are likely dependent on cell type and context we think this may have influenced our experimental outcomes and does not necessarily mean this interaction is not occurring in vivo. We have now included the following sentence into the discussion section:

“The inability of MFGE8 and CALR to induce a clear effect on fibroblasts might indicate that the suggested ligand-receptor interaction is not relevant in this particular setting in vitro, but does not necessarily exclude a role in vivo as the interactions are likely cell type and context dependent.” (line 443-446)

Reviewer 2

I recommend that the paper be accepted for publication with minor edits and clarification to the manuscript. The project is well conceived, executed, and presented, and is thorough. It applies a relatively new technique (scRNA_seq) to study an important pathology, IR injury, and wound healing in general.

We thank the reviewer for the positive and constructive feedback. We have tried to provide answers to all questions raised and incorporated suggestions to the manuscript to further improve clarity of the manuscript.

Minor questions/concerns:

- How many 384-well plates were processed per condition? I apologize if any of these information was already provided and I missed them.

We apologize for omitting this data. Per condition, we sequenced between 3-8 plates coming from 3-4 mice. Depending on the sequencing quality, some

conditions required more plates to be sequenced to have enough cells for reliable downstream analysis. This information has now been added to the methods section (line 588-589).

- Were more than one mice pooled for each condition? Were there biological replicate from separate litter?

Per condition, we used 3-4 mice from C57Bl6 background ordered from Charles River. The biological replicates were independently obtained from separate litters on different days. We have now mentioned that the biological replicates were independently obtained from separate litters in the methods section (line 589).

- Was there any corrections necessary for any batch effect? How were the datasets for different conditions integrated?

Per condition, we found that cells from the different batches were in general similarly distributed over all k-medoids clusters (Reviewers Figure 1 – 1dp Sham, 1dp IR and 14dp IR as an example). In addition, we found that cells from different batches were quite evenly dispersed in across the tSNE plot. Because we observed no to minimum batch effects within each condition, we reasoned that there should also be no to minimum batch effects between conditions. Therefore, we opted to not correct for this. We intergraded the cells from all conditions together by combining all UMIcount tables together, followed by random down sampling without replacement of reads in all cells to 1000 unique reads. These down sampled reads of cells from all conditions combined were than used for downstream clustering and tSNE plotting.

Reviewers Figure 1. K-medoids clusters and batch IDs of cells obtained per condition. a, tSNE plots showing the K-medoids cluster of each cell obtained from 1 day after sham (left), 1 day after IR (middle) or 14 days after IR surgery (right) separately. **b,** tSNE plots showing the batch ID of each cell obtained from 1 day after sham (left), 1 day after IR (middle) or 14 days after IR surgery (right) separately.

- Adult or terminally differentiated cardiomyocytes (CM) are often multi nucleation. Do the authors see any on facts? Were there any particular cluster that showed higher avg or median # of genes/transcripts that could be representative of these multi-nucleated CM?

Regrettably we were not able to clearly distinguish mono-nucleated from multi-nucleated cardiomyocytes by flow-cytometry. When we looked at the average total number of reads per cardiomyocyte cluster, we found that cluster 3 and to a certain extent cluster 4 had a substantially higher read count (Reviewers Figure 2). However, we are wary of concluding that these are the multi-nucleated cardiomyocytes. Recent (single-cell) RNA-sequencing studies comparing mono-nucleated with multi-nucleated cardiomyocytes in the healthy adult heart found similar gene expression profiles and average total read counts on whole-population level^{8,9}. Therefore, we think that the increased read counts in clusters 3 and 4 is indicative of higher total gene expression in these cells. However, this higher level of total gene expression might be a characteristic specific to these subtypes of cardiomyocytes independent of the total number of nuclei. To specifically look at the differences between mono- and multi-nucleated cardiomyocytes, we advise an approach similar as described by Windmueller et al.⁸, which uses labeled cardiomyocytes in combination with a nuclear staining to stratify mono-nucleated and multi-nucleated cardiomyocytes.

Reviewers Figure 2. Average total readcount per cell in cardiomyocyte clusters Bar graph showing average total reads of cells per cluster. Data are presented as mean +/- SEM. *** P<0.001, **** P<0.0001, Kruskal-Wallis test with Dunn's multiple comparison post-hoc test.

- On a related note, it is very difficult to distinguish a whole CM from broken CM. I suppose the DRAQ5 vs DAPI staining is helpful here. what was the aspect ratio of these cardiomyocyte cells (i.e., length along long axis vs. short axis)?

We agree with the reviewer that aspect ratio of the sorted cardiomyocytes would be of additional value. However, we unfortunately did not perform index-sorting during our FACS procedure, which disables the calculation of the cardiomyocyte aspect ratio's.

- Fig 1, clusters 6 (FB), 11 (NP) and 8,9 (MP) all seem like they have two clusters each. Broadly speaking, what are the clusters: IR vs sham? 1dp vs. 3dp? Or artifact from batch correction? (The individual cell-type clustering is too fine-grained to tell.) CM clusters 1,2,4,7 look fine in this respect. Could these cells have been used to integrate datasets?

In general, the subclusters in cluster 6 (FB) and clusters 8,9 (MP) are divided based on whether they are enriched in cells obtained from sham hearts or IR hearts (Reviewers Figure 3a-c). One subcluster consists for a substantial part of cells obtained from sham hearts while the other subcluster almost only consists of cells from IR hearts. For cluster 11, is less clear exactly what drives the subclustering. Almost all cells of cluster 11 are obtained from 1dp IR hearts, so subclusters are not specific for different conditions. However, since both subclusters contain cells from all batches of 1dp IR cells we do not think that this is a batch effect (Reviewers Figure 5d).

Recent studies have provided some evidence of different subsets of neutrophils present after myocardial infarction (extensively reviewed in ¹⁰). Therefore, we hypothesize that the subclusters in cluster 11 are potentially different subpopulations of neutrophils.

Reviewers Figure 3. Zoom in of cluster 6, 8, 9 and 11 in tSNE of all cells of all conditions showing condition or batch ID. a, tSNE map of all cells obtained from all conditions showing k-medoids clusters of cells. b, Zoom in of cluster 6 highlighting which cells were obtained from sham or IR hearts. c, Zoom in of cluster 8 and 9 highlighting which cells were obtained from sham or IR hearts. d, Zoom in of cluster 11 highlighting which cells were obtained from the different batches 1dp IR.

- For the Hypertrophic CM, do % MT-genes show any trend across time-points?
When comparing all cardiomyocytes in our dataset, we did find that the hypertrophic cardiomyocyte cluster (cluster 5) had a small but significant reduction in mitochondrial gene expression compared to the other cardiomyocytes (1,2,3,4) (Reviewers Figure 4a). When we specifically looked at

the cardiomyocytes in cluster 5 and stratify them based on the timepoint after IR, we could see a small decrease in mitochondrial gene expression in 14dp IR cardiomyocytes versus 3dp IR cardiomyocytes (Reviewers Figure 4b). However, this decrease failed to reach the significance threshold ($P=0.08$ – Mann-Whitney test). Therefore, with this data we cannot reliably conclude any difference in mitochondrial gene expression in the “hypertrophic” cardiomyocytes between different timepoints after IR. In humans, a reduced mitochondrial DNA content was observed in hypertrophic and failing cardiomyocytes compared to healthy cardiomyocytes (relative to nuclear DNA content)¹¹. This reduction in gene copies could explain the reduced mitochondrial gene expression in our hypertrophic cardiomyocytes (cluster 5) versus all other cardiomyocytes (clusters 1 to 4).

Reviewers Figure 4. Relative total mitochondrial gene expression across the cardiomyocyte clusters from Fig2. a, Relative mitochondrial gene expression across all non-hypertrophic cardiomyocytes (cluster 1,2,3,4 in Fig 2.) and in the hypertrophic cardiomyocytes (cluster 5 in Fig 2.) **b,** Relative mitochondrial gene expression in hypertrophic cardiomyocytes in cluster 5, which were obtained from 3dp IR and 14dp IR hearts. Data are presented as mean +/- SEM. *** $P<0.001$, Mann-Whitney test.

Are there any potential biases in cell-type recovery based on dissociation, e.g., 1dp vs 14dp IR? One would (perhaps naively) think that scar/fibrotic tissue may be harder to dissociate (in addition to the inherent challenges in dissociating cardiac tissue into single cell suspensions which is necessitating single nucleus RNA-seq in many studies)? What is cell viability % from each time-point? This may be calculated from the DAPI vs DRAQ5 fractions.

The reviewers correctly points out the possibility of bias in cell type recovery between different conditions or timepoints. The relative number of viable cells, indicated by DAPI-, does not significantly differ between the different conditions and timepoints (Supplementary Figure 1c). However, we cannot rule out that for instance the proportion of hypertrophic cardiomyocytes (Figure 2A, cluster 5), which are thought to flank the infarcted area, is underestimated because of increased digestion difficulty due to fibrotic tissue. Based on this feedback we

added Supplementary Figure 1c and implemented the following text into our discussion section:

“Although the relative number of viable cells, indicated by DAPI-, does not significantly differ between the different conditions and timepoints (Supplementary Fig. 1c), we cannot rule out that the proportion of hypertrophic cardiomyocytes is underrepresented because of increased digestion difficulty of fibrotic areas, where these cells are located (Figure 2A, cluster 5).”(line 413-417)

Supplementary Figure 1

Supplementary Figure 1. Cell sorting strategy yields viable cells with good membrane integrity from the adult heart. **a**, Percentage of DAPI negative cells after sorting and re-incubation with DAPI for several time points. **b**, Downstream analysis of sorted cells resulted in robust clustering shown by a bar graph depicting the Jaccard's similarity score for each cluster. As a rule of thumb, each cluster should have an index of > 0.6 to be robust. **c**, The fraction of Dapi negative cells as marker of viable cells for all different time points and conditions. Data are presented as mean +/- SEM.

- In figure 3e, how should we interpret 1dp IR and 3dp IR on two separate trajectories?

The reviewer correctly notes that there are 2 separate trajectories for the 1dp IR and the 3dp IR macrophages. Based on current literature, macrophages would be expected to gradually transition from a pro-inflammatory (M1) to an anti-inflammatory (M2) state as the wound-healing response progresses from the acute (1dp IR) to the intermediate (3dp IR) phase^{12,13}. This should result in a

trajectory where cells slowly transition from 1dp IR macrophages to 3dp IR macrophages, rather than forming 2 separate trajectories. Although this could have a biological explanation, we think it is more likely a consequence of the DDR tree algorithm. This pseudotime algorithm is designed to reconstruct a tree-like manifold over the data, resulting in a tree-like cell trajectory where cells with a different transcriptome are stratified in different branches¹⁴. This is an accepted model for differentiation programs for which this algorithm was originally developed, since these programs can often be represented by a tree-like graph. However, in the heart resident macrophages have recently been shown to have a more similar transcriptome to anti-inflammatory (M2) macrophages compared to pro-inflammatory (M1) macrophages⁵. The algorithm is not designed to reconstruct trajectories where macrophages are first transitioning from a resident state to a pro-inflammatory state, and the back to an anti-inflammatory state more similar to the initial resident state. We therefore decided in our manuscript to not draw biological conclusion from this trajectory, but to use it as a validation that resident, 1dpIR macrophages and 3dpIR macrophages predominantly separate from each other when using a different analysis approach. We reason that another type of manifold should be reconstructed to reliable study (and not just confirming) a biological process where cells gradually transition between different states.

- I really like the analyses, and the data are beautifully presented. Perhaps a bit more discussion to sum up the findings and highlight them in general terms? Some of the take-home messages may be made more general to make it more accessible to a wider community (I was underwhelmed reading the result and discussion sections, but the methods and figures were impressive. I particularly appreciated the ligand receptor analysis!)

In our revised manuscript better highlight the key conclusions and findings of our story.

- I don't quite see the utility of the iPSC-CM or fibroblast (FB) scratch experiments. It is not obvious from the optical images that the B2M treated FB heal better than the control. (Is 16 hr long enough to see this effect?) If there is one thing that the ligand receptor analysis shows, it is how ubiquitous communication among cell-types are. Perhaps a co-culture of CM and FB would have been more helpful and not just stimulating with certain factors/proteins. Even conditioned medium would be helpful. Anyway, the paper stands pretty well on its own without the in vitro experiments, and this section does not take anything away for it.

We would like to thank the reviewer for these constructive criticisms. Based on the received feedback we have changed the image of the B2M treated FB to clarify the healing effect of B2M. Also, the Western blots showing secretion of B2M by human cardiomyocytes have been removed to the supplementary data.

Reviewer 3

The manuscript by Molenaar et al describe the use of single-cell RNA-sequencing to understand the role of cardiac cell gene expression changes at 3 key time points following ischemic injury. The authors identify the expression and secretion of beta-2 microglobulin by cardiomyocytes following ischemic injury, which in turn activates

cardiac fibroblasts to promote scar formation. Overall, the manuscript is well written and organized. However, a few points would strengthen the overall impact of the paper.

We thank the reviewer for the positive feedback and suggestions, which we now used to improve our manuscript.

Major comments

1) In the myocardium, while cardiomyocyte encompass the greatest volume of cells, they are lower in numbers relative to other cell types, most notably endothelial and fibroblast cells. The data presented herein with sorting followed by scRNA-seq, suggest significant loss of both cell types. The authors should clarify how this may impact interpretation of the data.

The reviewer correctly observes a significantly larger cardiomyocyte population in our dataset than would be expected, while the endothelial cell and fibroblasts populations are smaller. This is a known limitation in our FACS-based sort-seq approach, which preferentially isolates larger cells like cardiomyocytes. This was previously highlighted in our paper describing the approach^{15,16}. We have now added the following section of text in the discussion to clarify how this might impact the interpretation of the data (line 459-467):

“A limitation of our approach is the inherent bias towards isolating larger cells defined by the gating parameters of the flow sorter^{17,18}. As a consequence, the relative abundance of cardiomyocytes in our dataset is higher than one would expect based on the cell-type composition of the heart (47% of cells in our dataset are cardiomyocytes, whereas approximately 30% of cells in the heart are cardiomyocytes)³³. As such, our dataset is less suitable to study the relative abundance of cell types in the heart during one timepoint. However, since the same inherent bias towards larger cells is present during all timepoints, relative changes in the abundance of a cell type between the different timepoints can be studied.”

2) Identification of subpopulations of cardiomyocytes in sham and ischemic myocardium is intriguing, but exploration into functional differences is limited (Fig. 2). The authors should include pseudotime plot analyses on cardiomyocytes, as shown in Figs. 3 & 4 to support their data.

*Per suggestion of the reviewer, we now have performed cell lineage trajectory (pseudotime) analysis on the cardiomyocytes population and added in to Figure 2 (Figure 2e) and Supplementary Figure 2 (shown below). We have also described our observation in results section **Ischemic injury induces a hypertrophy-associated gene program in a subset of cardiomyocytes** (line 216-227).*

“To further confirm transcriptomic heterogeneity in the cardiomyocytes, we additionally performed pseudotime and cell trajectory analysis on all cardiomyocytes using Monocle2²⁵ (Fig. 2e). Similar as in the tSNE plot, where cluster 1 to 3 did not form distinct clusters in tSNE space, cells from these clusters also did not form distinct branches but instead dispersed over multiple branches and interspersed with each (Fig. 2a,e). The branches containing most of the cells of cluster 1 to 3 showed comparable gradients of cardiomyocyte-marker expression as observed in the tSNE plot (Supplementary Fig. 2b-d). In contrast to cluster 1 to 3, cells from cluster 4 were

predominantly present in one branch (upper branch - Fig. 2e). Cell from cluster 5, the cluster with a hypertrophic transcriptional profile, were mainly aggregated on one location in the trajectory plot (around branch point 2 – Fig. 2e). Expression of genes enriched in cluster 5 was also highly expressed around this branch point (Supplementary Fig. 2f).”

Figure 2

Fig 2. Single-cell sequencing indicated cardiomyocyte heterogeneity. **a**, tSNE plot indicating transcriptomic similarities across cardiomyocytes obtained from all conditions. Different colors, symbols and numbers highlight the clusters as determined by k-medoids clustering of 1- Pearson’s correlation coefficient. **b**, Bar graph showing the proportion of cardiomyocytes originating from the different condition per cluster.. **c**, Heatmap showing expression of all genes significantly enriched in at least one cluster. Examples of genes enriched per cluster are depicted on the right side of the heatmap. **d**, Bubble plot of top GO-terms in gene ontology analysis on genes significantly enriched within the cluster. **(b-d)** Numbers highlighted in red depict cluster that mainly contains 3dp IR and 14dp IR cardiomyocytes (cluster 5). **e**, Cell trajectory analysis of cardiomyocytes, showing pseudotime on a color-coded scale (left) or highlighting the k-medoids cluster of each cardiomyocytes in the trajectory plot (right). The regions in the branches of the trajectory plot that contain mostly cells from k-medoids clusters 1,2,3 or cluster 4 or cluster 5 are highlighted by number.

Supplementary Figure 2

Supplementary Figure 2. tSNE analysis across cardiomyocyte populations indicate gradients of cardiomyocyte marker gene expression. **a**, Heatmap of the cell-to-cell transcriptome similarities (1 - Pearson's correlation coefficient) of 1033 cardiomyocytes obtained from all conditions combined. Cells are clustered based on transcriptome similarity using k-medoids clustering. Clusters identified through this method are also used in fig. 2. **b-f**, tSNE (left) and cell trajectory analysis (right) showing the distribution of expression of genes enriched in cluster 1 (**b**), cluster 2 (**c**), cluster 3 (**d**), cluster 4 (**e**) and cluster 5 (**f**). Expression is shown as normalized read count on a color-coded scale, with red depicting the highest expression and white lowest. Locations in the tSNE plot or in the trajectory plot that contain mostly cells from a cluster are highlighted by numbers of that respective cluster.

3) Macrophages have been well-characterized to have both resident and infiltrating macrophage populations. Based on the current literature, how do your macrophage subpopulations (Fig. 3) compare to those that have been published? The authors should stratify their macrophage subpopulations to align with either a resident or infiltrating macrophage phenotype.

The reviewer correctly highlights that resident and infiltrating macrophages after MI have recently been well characterized on single-cell level^{5,18}. The infiltrating macrophages are the macrophages of the 1dpIR- and 3dpIR-enriched clusters. As stated in the results, they express pro-inflammatory genes for 1dp IR macrophages and pro-wound healing genes for 3dp IR macrophages. The genes enriched in the 1dpIR macrophages were also found in pro-inflammatory macrophages (M1 MΦ) in the study by Farbehi et al.⁵, and include genes like Spp1, Arg1, Ccl7, Ccl2, Pf4 (Database S3). In addition, the genes enriched in the 3dpIR macrophages were also found in the anti-inflammatory macrophages (M2 MΦ) in the study by Farbehi et al.⁵, and include genes like Cd74, Apoe and Ms4a7(Database S3). However, the reviewer accurately notes that we did not distinctly highlight the resident macrophages. We have now highlighted in the result section that macrophages from cluster 1 most likely constitute the resident macrophages (line 252-257, text shown below). These macrophages come from all conditions, and genes found highly expressed in resident macrophages by Farbehi et al.⁵ were also highly expressed in this cluster (Supplementary Figure 4). We have added Supplementary Figure 4 clearly showing higher expression of most genes found enriched in resident macrophages by Farbehi et al.⁵ to make a clear distinction between the resident (cluster 1) and infiltrating (1dp IR and 3dp IR cluster) macrophages.

Implemented textual changes line 252-257:

“We also observed a cluster containing a substantial number of macrophages originating from sham hearts and from hearts during the intermediate and chronic phases of wound healing (cluster 1) (Fig. 3b). Genes that were previously found enriched in resident cardiac macrophages were also higher expressed in this cluster (Fig. 3c, Supplementary Fig. 5, Data Set 3)²⁷. We therefore concluded that cells from cluster 1 most likely constitute the resident macrophages. “

Supplementary Figure 4

Supplementary Figure 4. Bar graph showing fold change of genes previously shown to be enriched in resident cardiac macrophages by single-cell RNA sequencing. Bar graph depicting the fold change of genes in each macrophage cluster compared to all other clusters. Gene were previously found to be enriched in resident cardiac macrophages in a single-cell RNA sequencing study by Ferbahi et al.²⁷. *P<0.05, calculated using binomial counting statistics of the RaceID2 algorithm.

4) The expression of B2M increases at d3 relative to d1 post-MI (Fig. 6A). Does the expression of B2M (and its cognate receptors) change if you perform the same analyses in Fig.5 on 3d post-MI samples? Since these data are available, these analyses would strengthen the mechanism.

Per suggestion of the reviewer, we have analyzed which ligands would still be selected 3dp IR, since the secretion of B2M increases 3dp IR. We observed that the number of selected ligands 3 dp IR was much smaller compared to 1dp IR (Reviewers Figure 5a) This suggests that cardiomyocytes during the intermediate phase of wound healing could be less involved in intercellular communication. We next specifically focus on the three factors that we investigate further in Figure 6. Broadly speaking, we can see that changes in transcript levels precede changes in ligand secretion into the serum. For CALR, the higher expression at transcript level in cardiomyocytes is 1dp IR (Reviewers Figure 5b) and secretion is also highest at this timepoint (Reviewers Figure 5c-d). However, transcript levels are lower in 3dp IR cardiomyocytes compared to sham cardiomyocytes but secretion of CALR is not (yet) lower compared to sham but at the same level. For both MFGE8 and B2M, the highest transcript level is 1dp IR, with upregulation going down again 3dp IR (back to baseline for B2M). However, secretion of MFGE8 and B2M is higher at 3dp IR compared to 1dp IR. We reason that changes in transcript levels precede changes in protein secretion levels due to the general delay in protein synthesis/decay, which

cause a temporal delay between induction/repression of transcription of a gene and corresponding changes at the protein level¹⁹.

Reviewers Figure 5. Expression of ligands in 1dp IR vs 3dp IR cardiomyocytes a Venn-diagram of all ligands that are selected in 1dp IR and 3dp IR cardiomyocytes using the filtering strategy described in Fig. 5A, but before exclusion of candidates that were part of the ECM of highly expressed in other cell types. **b**, Changes in expression of the 3 ligands that were further studied in Fig 5 and Fig 6 in 1dp IR and 3dp IR cardiomyocytes relative to sham cardiomyocytes. Data is shown and the log2 of the fold change relative to sham cardiomyocytes. The dotted line represents an upregulation of a fold change of 3, which is required for inclusion of the ligand into the screening strategy. Data are presented as mean +/- SEM. * P<0.05, ** P<0.01, *** P<0.001, **** P<0.0001 compared to sham, Wald test with Benjamini-Hochberg correction. **c**, Immunoblots on mouse plasma at different time points post-IR or 1 day post-sham (adapted from Fig 6.A). **d**, Quantification of immunoblots. * P<0.05 compared to sham, Kruskal-Wallis test with Dunn's multiple comparison post-hoc test.

5) It is unclear how post-MI induced B2M expression, which is shown to increase in murine cardiomyocytes by scRNA-seq (Fig. 5) and cardiac tissue by WB (Fig. 6A), is localized to infiltrating immune cells by histology (Fig. 6A). Do macrophages play a significant role in B2M expression and modulation of fibroblast activity? This should be more clearly explored with additional supporting data.

We thank the reviewer for this comment. The reviewer correctly observes expression of B2M by other cell types than cardiomyocytes. Based on this feedback and comment 7 we have now included histological images to clarify that in addition to CM, also MP, NP, EC and FB express B2M (Supplementary Figure 11). Our analysis reveals many potential intercellular communications to overlap between several cell types (Data Set 9) and as highlighted in our results section, the only proteins in our cellular interaction network that were uniquely secreted by cardiomyocytes are NPPA and NPPB. We would like to emphasize that our data suggests that in addition to other cell types, cardiomyocytes contribute to fibroblast activation during the acute phase post-IR. To clarify this, we made the following two changes in our manuscript.

1. *We have implemented the text below in our discussion section (line 446-453)*
“Additionally, it should be noted that the identified ligand-receptor interactions are often not unique between 2 specific cell types. Different cell types can often co-act to enable ligand-receptor interactions. This also appears to be the case for B2M as B2M was also expressed in other cardiac cell types than cardiomyocytes (Supplementary figure 11, Data Set 9). Based on our data we know that increased B2M expression from cardiomyocytes contributes to fibroblast activation during the acute phase post IR, however, this does not exclude other cell types to stimulate fibroblasts via B2M as well.”

Many molecular interactions in the cardiac interactome as revealed here are not unique to a specific cell type combination. This suggest that different cell types often co-act to enable ligand-receptor interactions. This seems also to be the case for B2M as in our dataset only NPPA and NPPB were uniquely secreted ligands by cardiomyocytes. Indeed, the other cardiac cell types were also found to express B2M (Supplementary Figure 11, Data Set 9). Therefore, our data suggest that increased b2m expression from cardiomyocytes contributes to fibroblast activation during the acute phase post IR. However, this does not exclude other cell types to stimulate fibroblasts via B2M as well.

2. *We changed the summary sentence of the results section (line 365-368). It now mentions that cardiomyocytes contribute to fibroblast activation:*
“These results reveal the potential intercellular communication networks between all main cardiac cell types during each phase of infarct healing and suggest that increased secretion of B2M by stressed cardiomyocytes post-IR, contributes to the activation of fibroblasts.”

Supplementary Figure 11

a

b

Supplementary Figure 11. B2M expression per cell type. **a**, Normalized B2m reads per cell type. CM; cardiomyocytes, FB; fibroblasts, MP; macrophages EC; endothelial cells, NP; neutrophils. **b**, Images captured by fluorescent microscopy showing expression of B2M by non-cardiomyocyte cells in the heart. All 1dp IR images were taken from the border zone. Cell markers used were Vimentin (FB), CD68 (MP), CD31 (EC), NIMP-R14 (NP). Scale bar, 25 μ m. white arrows indicate cells that show expression of B2M and expression of the corresponding cell markers.

6) All immunoblots in Fig. 6 should be quantitatively analyzed and included in the figure.

Based on this suggestion we have now included quantifications of the immunoblots in Figure 6.

Figure 6

Fig 6. B2M is secreted by stressed cardiomyocytes to activate fibroblasts **a**, Western blot on mice plasma at different time points post-IR or 1 day post-sham. Equal loading of plasma proteins was confirmed by Coomassie staining (Supplementary Fig. 9) **b**, Quantification of immunoblots in **a**, * $P < 0.05$ compared to sham, Kruskal-Wallis test with Dunnett's multiple comparison post-hoc test. **c**, cognate receptor expression of selected ligands in NIH/3T3 fibroblasts is shown by RT-qPCR. **d**, Upregulation of two myofibroblast (*Acta2* and *Vim*) markers in NIH/3T3 cells after treatment with B2M, TGF- β served as positive control ($n = 9$ from 3 independent experiments). **e**, Increased wound healing of NIH/3T3 cells after treatment with B2M, assessed by scratch assay. Images show representative pictures for each condition. Dashed lines indicate gap size. **f**, Quantification of wound healing assay relative to corresponding control ($n = 6$ from 2 independent experiments). Scale bars, 200 μ m. **g**, Immunohistochemistry images of heart slices after IR or sham surgery. White arrows indicate cardiomyocytes (larger cells with diminished CTNT staining) within the infarct zone that show B2M expression. CTNT, cardiac troponin T. Scale bars, 100 μ m (left and middle panels) and 20 μ m (right panels). Data are presented as mean \pm SEM. * $P < 0.05$, *** $P < 0.001$. One-way ANOVA with multiple comparison vs control and Dunnett's adjustment for multiple hypothesis testing.

7) Histology in Fig. 6g should be analyzed with additional cell-specific antibodies to delineate the expression of B2M within each cell type described herein (i.e., cardiomyocytes, fibroblasts, macrophages, neutrophils, and endothelial cells), as alluded to in comment 5. Quantitative analyses of B2M expression in each cell type should be included at d1 and d3 post-MI.

Based on this feedback and comment 5 we have now included histological images to clarify that in addition to CM, also MP, NP, EC and FB express B2M (Supplementary Figure 11, shown under major comment 5). As the images are not suitable for a reliable quantification, we can unfortunately not provide a quantification based on histological analysis. However, we did include additional cell-specific transcriptomic data of B2M (Supplementary Figure 11a) which shows that compared to CM, other cell types show comparable or slightly higher b2m expression at 1dpIR. Notably, CM are the only cell type that show an increased b2m expression during the acute phase post-IR compared to sham (Supplementary Figure 11a, shown under Major comment 5).

Minor Comments

1) Fig. 1 e-g. You describe a cell population with unknown designation (3; Fig. 1e-f). How does population 3 shift at each time point, as you've shown with the other cell populations (Fig. 1g)?

This unknown cluster (cluster 3) constitutes cells from all conditions. We have now added this cluster to the bar graph (Figure 1g) in our manuscript.

Fig 1g.

2) Clarify heatmap key (Right part of pseudotime plots – Figs. 3 & 4), which reflects the colour-coding from tSNE plots (Figs. 3 & 4)

Thank you for the suggestion. We have now clarified the heatmap key belonging to the trajectory plots.

Figure 3e.

Figure 4e

3) Since the iPS data (Fig. 6b) does not align with the murine model, it should be placed in the supplement.

Based on this feedback we have now removed these data to the supplement. (see Figure 6 shown under Major comment 6)

4) Fig. 6e-f. The order and labeling in which these figures are placed is confusing. The quantitative measurements defining the change in area (Fig. 6f) should be placed after the representative images (Fig. 6e).

Thank you for the suggestion. We have now adapted this. (see Figure 6 shown under Major comment 6)

References

- 1 Kovacic, J. C. *et al.* Endothelial to Mesenchymal Transition in Cardiovascular Disease: JACC State-of-the-Art Review. *J Am Coll Cardiol* **73**, 190-209, doi:10.1016/j.jacc.2018.09.089 (2019).
- 2 Li, R. & Guan, M. X. Human mitochondrial leucyl-tRNA synthetase corrects mitochondrial dysfunctions due to the tRNA^{Leu}(UUR) A3243G mutation, associated with mitochondrial encephalomyopathy, lactic acidosis, and stroke-like symptoms and diabetes. *Mol Cell Biol* **30**, 2147-2154, doi:10.1128/mcb.01614-09 (2010).
- 3 Singh, S. *et al.* Genome Wide Analysis Approach Suggests Chromosome 2 Locus to be Associated with Thiazide and Thiazide Like-Diuretics Blood

- Pressure Response. *Sci Rep* **9**, 17323, doi:10.1038/s41598-019-53345-5 (2019).
- 4 Saeki, M. *et al.* Monad, a WD40 repeat protein, promotes apoptosis induced by TNF-alpha. *Biochem Biophys Res Commun* **342**, 568-572, doi:10.1016/j.bbrc.2006.02.009 (2006).
- 5 Farbehi, N. *et al.* Single-cell expression profiling reveals dynamic flux of cardiac stromal, vascular and immune cells in health and injury. *Elife* **8**, doi:10.7554/eLife.43882 (2019).
- 6 Jablonski, K. A. *et al.* Novel Markers to Delineate Murine M1 and M2 Macrophages. *PLoS One* **10**, e0145342, doi:10.1371/journal.pone.0145342 (2015).
- 7 Mouton, A. J. *et al.* Mapping macrophage polarization over the myocardial infarction time continuum. *Basic Res Cardiol* **113**, 26, doi:10.1007/s00395-018-0686-x (2018).
- 8 Windmueller, R. *et al.* Direct Comparison of Mononucleated and Binucleated Cardiomyocytes Reveals Molecular Mechanisms Underlying Distinct Proliferative Competencies. *Cell Rep* **30**, 3105-3116 e3104, doi:10.1016/j.celrep.2020.02.034 (2020).
- 9 Yekelchik, M., Guenther, S., Preussner, J. & Braun, T. Mono- and multi-nucleated ventricular cardiomyocytes constitute a transcriptionally homogenous cell population. *Basic Res Cardiol* **114**, 36, doi:10.1007/s00395-019-0744-z (2019).
- 10 Puhl, S. L. & Steffens, S. Neutrophils in Post-myocardial Infarction Inflammation: Damage vs. Resolution? *Front Cardiovasc Med* **6**, 25, doi:10.3389/fcvm.2019.00025 (2019).
- 11 Pisano, A. *et al.* Impaired mitochondrial biogenesis is a common feature to myocardial hypertrophy and end-stage ischemic heart failure. *Cardiovasc Pathol* **25**, 103-112, doi:10.1016/j.carpath.2015.09.009 (2016).
- 12 Forte, E., Furtado, M. B. & Rosenthal, N. The interstitium in cardiac repair: role of the immune-stromal cell interplay. *Nat Rev Cardiol* **15**, 601-616, doi:10.1038/s41569-018-0077-x (2018).
- 13 Frangogiannis, N. G. The inflammatory response in myocardial injury, repair, and remodelling. *Nat Rev Cardiol* **11**, 255-265, doi:10.1038/nrcardio.2014.28 (2014).
- 14 Qiu, X. *et al.* Reversed graph embedding resolves complex single-cell trajectories. *Nat Methods* **14**, 979-982, doi:10.1038/nmeth.4402 (2017).
- 15 Gladka, M. M. *et al.* Single-Cell Sequencing of the Healthy and Diseased Heart Reveals Cytoskeleton-Associated Protein 4 as a New Modulator of Fibroblasts Activation. *Circulation* **138**, 166-180, doi:10.1161/CIRCULATIONAHA.117.030742 (2018).
- 16 Molenaar, B. & van Rooij, E. Single-Cell Sequencing of the Mammalian Heart. *Circ Res* **123**, 1033-1035, doi:10.1161/CIRCRESAHA.118.313531 (2018).
- 17 Pinto, A. R. *et al.* Revisiting Cardiac Cellular Composition. *Circ Res* **118**, 400-409, doi:10.1161/CIRCRESAHA.115.307778 (2016).
- 18 Dick, S. A. *et al.* Self-renewing resident cardiac macrophages limit adverse remodeling following myocardial infarction. *Nat Immunol* **20**, 29-39, doi:10.1038/s41590-018-0272-2 (2019).
- 19 Liu, Y., Beyer, A. & Aebersold, R. On the Dependency of Cellular Protein Levels on mRNA Abundance. *Cell* **165**, 535-550, doi:10.1016/j.cell.2016.03.014 (2016).

REVIEWERS' COMMENTS:

Reviewer #2 (Remarks to the Author):

The project is well-conceived, executed and thorough. It applies a relatively new technique, scRNA-seq to study an important pathology- IR injury, and wound healing in general, and thus would be of great interest to a broad community. I find that the authors have addressed all reviewers' comments satisfactorily. I recommend that the paper be accepted for publication.

Reviewer #4 (Remarks to the Author):

the authors have adequately addressed reviewers' concerns. I have no further critiques.

Reviewer #5 (Remarks to the Author):

In my opinion, judging by their comments, Reviewer 3 was fairly positive to begin with. The major comments were hardly destructive and were feasible & easy to reply. Overall, the authors did a very good job addressing them.

Regarding major comment 4: Please include these data in the manuscript and include your reasoning and explanations.

Regarding major comment 7: The reply to this comment is unsatisfactory. The authors must provide better quality images that will allow delineating the expression of B2M within each cell type described herein (i.e., cardiomyocytes, fibroblasts, macrophages, neutrophils, and endothelial cells) as previously demanded as well as a proper quantification of B2M expression in each cell population. The presented IF is of too low quality to address the issue raised here.